

**Modelling the Impacts of Iodine Chemistry on the Northern Indian Ocean Marine**
**Boundary Layer**
Anoop S. Mahajan[1*], Qinyi Li[2], Swaleha Inamdar[1,3], Kirpa Ram[3], Alba Badia[4] and Alfonso
Saiz-Lopez[2]
[1]Indian Institute of Tropical Meteorology, Ministry of Earth Sciences, Pune, 411016, India
[2]Department of Atmospheric Chemistry and Climate, Institute of Physical Chemistry
Rocasolano, CSIC, Madrid, 28006, Spain
[3]Institute of Environment and Sustainable Development, Banaras Hindu University, Varanasi,
221 005, India
[4]Institute of Environmental Science and Technology (ICTA), Universitat Autònoma de
Barcelona (UAB), Barcelona, Spain
* Corresponding author: Anoop S. Mahajan (anoop@tropmet.res.in); phone: +91 20 2590 4526





**Abstract**
Recent observations have shown the ubiquitous presence of iodine oxide (IO) in the Indian
Ocean marine boundary layer (MBL). In this study, we use the Weather Research and
Forecasting model coupled with Chemistry (WRF-Chem version 3.7.1), including halogens
(Br, Cl and I) sources and chemistry, to quantify the impacts of the observed levels of iodine
on the chemical composition of the MBL. The model results show that emissions of inorganic
iodine species resulting from the deposition of ozone ($O_3$) on the sea surface are needed to
reproduce the observed levels of IO, although the current parameterisations overestimate the
atmospheric concentrations. After reducing the inorganic emissions by 40%, a reasonable
match with cruise-based observations is found. A strong seasonal variation is also observed,
with lower iodine concentrations predicted during the monsoon period when clean oceanic air
advects towards the Indian subcontinent, and higher iodine concentrations predicted during the
winter period, when polluted air from the Indian subcontinent increases the ozone
concentrations in the remote MBL. The results show that significant changes are caused by the
inclusion of iodine chemistry, with iodine catalysed reactions leading to regional changes of
up to 25% in $O_3$, 50% in nitrogen oxides (NO and $NO_2$), 15% in hydroxyl radicals (OH), 25%
in hydroperoxyl radicals ($HO_2$), and up to a 50% change in the nitrate radical ($NO_3$). Most of
the large relative changes are observed in the open ocean MBL, although iodine chemistry also
affects the chemical composition in the coastal environment and over the Indian subcontinent.
These results show the importance of including iodine chemistry in modelling the atmosphere
in this region.
**Keywords:** iodine, northern Indian Ocean, marine boundary layer, oxidising capacity





## 1. Introduction

Iodine compounds, emitted from the ocean surface, have been implicated in causing changes to the chemical composition of the marine boundary layer (MBL (Carpenter, 2003; Platt and Honninger, 2003; Saiz-Lopez et al., 2012a; Saiz-Lopez and von Glasow, 2012; Simpson et al., 2015). The known effects include changes to the oxidising capacity through the depletion of ozone ($O_3$) (Iglesias-Suarez et al., 2018; Mahajan et al., 2010b; Read et al., 2008; Saiz-Lopez et al., 2007) changes to the hydrogen oxides ($HO_x = OH$ & $HO_2$) and nitrogen oxides ($NO_x = NO$ and $NO_2$) concentrations (Bloss et al., 2005; Chameides and Davis, 1980) and oxidation of mercury (Wang et al., 2014). Coastal emissions of iodine compounds, through known biogenic sources such as macroalgae, have been shown to contribute significantly to new particle formation (McFiggans, 2005; O'Dowd et al., 2002, 2004). It has been suggested that even in the open ocean environments with low iodine emissions, it can participate in new particle formation (Allan et al., 2015; Baccarini et al., 2020; Sellegri et al., 2016). Recent ice-core observations in the high altitude Alps in Europe and in Greenland have shown an increase in the atmospheric loading of iodine compounds, which highlights the importance of understanding iodine cycling for accurate future projections (Cuevas et al., 2018; Legrand et al., 2018).

Over the last two decades, several field campaigns have focused on the measurement of iodine oxide (IO), which can be used as a proxy for iodine chemistry in the MBL. These observations made across the world show a near-ubiquitous presence of IO across the Pacific, Atlantic, and Southern Oceans with concentrations reaching as high as ~3 parts per trillion by volume (pptv) in the open ocean environment (Alicke et al., 1999; Allan et al., 2000; Commane et al., 2011; Furneaux et al., 2010; Gómez Martín et al., 2013; Großmann et al., 2013; Mahajan et al., 2012, 2009, 2010a, 2010b, 2011; Platt and Janssen, 1995; Prados-Roman et al., 2015; Read et al., 2008; Saiz-Lopez and Plane, 2004; Seitz et al., 2010; Stutz et al., 2007; Wada et al., 2007;





Zingler and Platt, 2005). Until recently, the Indian Ocean was the most under-sampled region
for iodine species, but cruises that were a part of the Indian Southern Ocean Expeditions
(ISOEs) and the International Indian Ocean Expedition- 2 (IIOE-2) have confirmed the
presence of up to 1 pptv of IO in this region's MBL (Inamdar et al., 2020; Mahajan et al.,
2019a, 2019b).
Over the Indian Ocean, intense anthropogenic pollution from Southeast Asia mixes with
pristine oceanic air. The mixing of polluted continental and clean oceanic air masses results in
unique chemical regimes, which change drastically due to distinct seasonal circulation patterns,
such as the seasonally varying monsoon. During the winter monsoon season (November to
March), high pollution levels are regularly observed over the entire northern Indian Ocean
(Lelieveld et al., 2001), while during the summer monsoon (June-September), clean air
dominates the atmospheric composition, leading to distinct chemical regimes (Lawrence and
Lelieveld, 2010). For the other transitional months, especially the pre-summer monsoon period
(March-June), the offshore pollution is in general weaker compared to the winter monsoon
conditions (Sahu et al., 2006). The changing atmospheric composition over the Indian Ocean
can interact with oceanic biogeochemical cycles and impact marine ecosystems, resulting in
potential feedbacks. This is indeed the case of inorganic iodine emissions (hypoiodous acid,
HOI and molecular iodine, $I_2$), which are considered to be the major sources of reactive iodine
species from the ocean surface (Carpenter et al., 2013; MacDonald et al., 2014). The emission
of both species depends on the deposition of atmospheric $O_3$, which shows a strong seasonal
cycle due to the changes in the composition of the overlying airmasses. However, even though
the emission of iodine compounds is expected to increase during higher pollution periods,
anthropogenic $NO_x$ can lead to titration of iodine in the atmosphere, leading to the formation
of the relatively stable iodine nitrate ($IONO_2$), which effectively reduces the impact of iodine



on the atmosphere in terms of ozone depletion and also new particle formation (Mahajan et al.,
2009, 2011, 2019b).
Recent modelling studies have made an attempt at quantifying the impact of iodine on a global
scale (Saiz-Lopez et al., 2012b, 2014; Sherwen et al., 2016; Stone et al., 2018) and at regional
scales (Li et al., 2019, 2020; Muñiz-Unamunzaga et al., 2018; Sarwar et al., 2015). Although
both approaches have shown significant effects of iodine on the atmosphere, a strong difference
is observed in different regions due to the existing chemical regimes. Amongst the regional
studies, estimates in the eastern Pacific using the Weather Research and Forecasting model
coupled with Chemistry (WRF-Chem) suggest that halogens account for about 34% of the total
ozone depletion in the MBL, of which iodine compounds cause about 16% (Badia et al., 2019).
In China, the contribution of iodine to the halogen-mediated effect on atmosphere oxidising
capacity has been calculated to be up to 29% (Li et al., 2020). Using the Community Multiscale
Air Quality Model (CMAQ), Li et al. (Li et al., 2019) showed that combined halogen chemistry
(chlorine, bromine and iodine) induces variable effects on OH (ranging from -0.023 to 0.030
pptv) and $HO_2$ (in the range of -3.7 to 0.73 pptv), reduces nitrate radical ($NO_3$) concentrations
(~20 pptv) and $O_3$ (by as much as 10 ppbv), decreases $NO_2$ in highly polluted regions (by up
to 1.7 ppbv) and increases $NO_2$ (up to 0.20 ppbv) in other areas. Another study using the same
model suggested that in the northern hemisphere, halogen chemistry, without higher iodine
oxides photochemical breakdown, leads to a reduction of surface ozone by ~15%, whereas a
simulation including their breakdown leads to reductions of ~48% (Sarwar et al., 2015).
However, studies are lacking in the quantification of the impact of iodine over the Indian Ocean
MBL. Here, we study the effects of iodine chemistry on the atmospheric composition in the
northern Indian Ocean MBL, a region where effects of iodine have not been studied hitherto,
using the WRF-Chem model over three different periods in a year. We explore the geographical



and seasonal variability through quantification of iodine-mediated changes in ozone, $HO_x$ and
$NO_x$.

**2. Methodology**
The WRF-Chem model (version 3.7.1), which included a full halogen scheme (Cl, Br, and I)
was used in the present study. The halogen chemistry scheme used in WRF-Chem and a
detailed description of the model setup are described in past studies (Badia et al., 2019; Li et
al., 2020). Sources of reactive iodine species considered in this study are an oceanic source of
organic iodine compounds ($CH_2ICl$, $CH_2IBr$, $CH_2I_2$, and $CH_3I$) and inorganic compounds from
the ocean surface (HOI and $I_2$). The sea-to-air fluxes of organic compounds were calculated
online (Liss and Slater, 1974). The oceanic emission of inorganic iodine (HOI and $I_2$), which
is dependent on the deposition of $O_3$ to the surface ocean and reaction with iodide ($I^-$) was
calculated online using a parameterisation based on Badia et al. (2019), which was computed
using the empirical laboratory measured parameterisations by Carpenter et al. (2013) and
MacDonald et al. (2014). These emissions produced much higher than observed levels of IO
in the northern Indian Ocean MBL. The reasons for the overestimation are discussed further in
Section 3.2. Hence for the rest of the analysis, the emissions of $I_2$ and HOI were reduced by
40% (i.e. 60% of the standard emission parameterisation).
The domain for the simulations was selected to cover the Indian subcontinent and the northern
Indian Ocean (as shown in Figure 1). We used a spatial resolution of 27 km and 30 vertical
layers (sigma levels of 1.00, 0.99, 0.98, 0.97, 0.96,0.95, 0.94, 0.93, 0.92, 0.91,0.89, 0.85, 0.78,

134    0.70, 0.60,0.51, 0.43, 0.36, 0.31, 0.27,0.23, 0.20, 0.17, 0.14, 0.11,0.08, 0.05, 0.03, 0.02, 0.01,

0.00) with the surface layer ~20 m above ground level and 10 layers within the boundary layer.
The simulation period included three seasons in the year of 2015 (pre-monsoon in April;





137 summer monsoon in July; and the winter monsoon period in January). We ran the WRF-Chem

138 model for the months of January, April and July with an extra spin-up period of 15 days. The

139 reason for choosing these three months is the different chemical regimes that result over the

140 Indian Ocean due to changing meteorological conditions. Figure 1 shows the monthly averaged

141 wind direction and speed over the northern Indian Ocean, which shows the drastic differences

142 in air masses over the three seasons. Using these considerations, three sets of simulations were

143 conducted. The BASE scenario considered no iodine emissions from the ocean surface; the

144 orgI scenario considered only emissions of organoiodides as mentioned above; and the HAL

145 scenario considered emissions of both inorganic iodine and the organoiodides. Changes in

146 atmospheric compositions between BASE and HAL represent the impact of the overall iodine

147 sources and chemistry; while those between the BASE and orgI scenarios represent the impact

148 of organic iodine emissions; and the difference between orgI and HAL shows the impact of

149 inorganic emissions of iodine from the ocean surface (HOI and $I_2$).

150 The model results were validated using observations from cruise-based campaigns in the Indian

151 Ocean, i.e. during the 2[nd] International Indian Ocean Expedition (December 2015) and the 8[th]

152 Indian Southern Ocean Expedition (ISOE-8) (January 2015) (Mahajan et al., 2019b, 2019a).

153 Unfortunately, observations were available only during the winter monsoon period, and hence

154 no direct validation was possible during the other seasons. Observations of IO in the MBL,

155 along with surface ozone concentrations, were used for the validation of the model simulations.

156 The MBL in the model results was defined as the lowest 10 layers (1.0 km above sea level).

157 The domain chosen for the model simulations, along with the tracks of the cruises from which

158 data was used for validation are shown in Figure 2.

159

160 **3. Results and Discussions**





### 3.1 Model validation

A comparison between the model simulated IO and $O_3$ from the HAL scenario and observations made during the IIOE-2 and 8[th] ISOE expeditions is shown in Figure 3. The top panel shows a comparison between the modelled and observed IO mixing ratios, with both the model and observations showing IO values below 1 pptv for all the locations. For most of the data points, the model simulated IO is slightly higher than the observations, although within the uncertainty. It should be remembered that this close match is after reducing the emissions of inorganic iodine species from the ocean surface by 40% (discussed further is Section 3.2). The largest mismatch is observed close to 5° N, where the model predicts approximately 0.9 pptv, while the observations show a low of 0.23±0.16 pptv. A point to note is that the IIOE-2 observations were from December 2015, and hence an exact match is not expected. Nonetheless, the comparison shows that the model does a good job at reproducing the levels of IO observed in the northern Indian Ocean. The levels observed and simulated IO are similar to the west Pacific (Badia et al., 2019) and in the South China sea (Li et al., 2020) but lower than the modelled and observed values of ~1.5 pptv in the tropical Atlantic MBL (Mahajan et al., 2010b).

The bottom panel of Figure 3 shows a comparison between the model simulated $O_3$ with the observations. Although the match between the model and observations is good in the northern Indian Ocean, there is a mismatch in the open ocean closer to the equator, with the model predicting higher values than the observations. The decreasing trend towards the open ocean is well captured by the model, with higher values observed close to the Indian coast where larger anthropogenic emissions are present. The average overestimation of ozone across all the locations where observations were available was ~25%. The model captures well the difference between the IIOE-2 and the ISOE-8 cruises, which started from the west and east coasts of India, respectively. Larger values of $O_3$ were observed during the ISOE-8, which were seen in



the model simulations too, and shows that the $O_3$ concentrations during the winter months are
higher to the east of India as compared to the west.
**3.2 Geographical distribution of IO**
Figure 4 shows the geographical distribution of daytime averaged IO across the selected
domain during the three seasons, for the orgI and HAL cases, along with the difference between
the two. The orgI scenario shows significantly low values across the domain, with a peak of
only 0.06 pptv (in January) in the western Indian Ocean close to the equator (Figure 4, top
panels). When averaged across the whole domain for the boundary layer, the mean IO mixing
ratio is a negligible 0.011±0.009 pptv in January, 0.008±0.006 pptv in April and 0.012±0.009
pptv in July (Table 1). Even if only the MBL is considered after applying a land mask, the
mean IO mixing ratio is only 0.015±0.009 pptv in January, 0.011±0.006 pptv in April and
0.015±0.008 pptv in July (Table 1). At such low concentrations, iodine chemistry would not
have any measurable impact on atmospheric chemistry. The values closer to the Indian
subcontinent are negligible, although a high of ~0.04 pptv is seen close to the western Indian
and Pakistani coast during the summer monsoon period (July). It is well known that this region
experiences strong mixing in the northern Arabian sea during the summer monsoon period,
which triggers plankton blooms resulting in high productivity (Qasim, 1982). For the current
model runs, emissions of organic iodine are based on a climatology concentration of organic
halogens in the sea water (Ziska et al., 2013), which show high organoiodides emissions in this
region. However, despite the being an area of high productivity, the values of IO predicted in
the orgI scenarios are significantly lower than the observations (by a factor of 10-20; Figure 2)
and show the need for an inorganic iodine flux. Such a flux has been suggested to be ubiquitous
and dependent on the ozone deposition and seawater iodide concentrations (Carpenter et al.,
2013; MacDonald et al., 2014).



The middle panels in Figure 4 show the distribution of IO for the HAL scenario, which includes
an inorganic iodine flux of $I_2$ and HOI as mentioned earlier. The flux strength has been reduced
by 40% (i.e. 60% of the standard emission parameterisation) compared to the past studies to
get a closer match between the observations and the model. Without such a reduction, the
model predicts a peak of ~1.7 pptv in the domain, which is almost double the peak value
observed during the IIOE-2 or ISOE-8, even when the uncertainty in the observations is
considered (Figure 2). The main drivers for a sea-to-air flux of HOI and $I_2$ are the concentration
of iodide in the seawater and the atmospheric ozone concentrations. The seawater iodide
concentration in the model was estimated using the MacDonald et al. (2014) parameterisation,
which is based on the sea surface temperature. This is also the largest uncertainty in the
inorganic iodine emissions parameterisation. Recent studies have shown that the MacDonald
et al. parameterisation underestimates the seawater iodide in the Indian Ocean (Inamdar et al.,
2020), with the model predicted mean iodide in the domain being 117.4±1.4 nM  (range: 113
to 119 nM). Ship-based observations in the same region show a mean iodide value of
185.8±66.0 nM (range of 100 to 320 nM) (Chance et al., 2019, 2020). Iodide observations were
unfortunately not made during the same time as the model runs, but it is unlikely that the model
overestimates the iodide considering the range of reported observations. Indeed, if we use the
mean observed values for the seawater iodide concentrations, the $I_2$ flux would increase by
~58% and HOI flux would increase by 44%, rather than both decrease by 40% as necessary.
The second reason for overestimating the sea-to-air iodine flux could be the overestimation of
ozone in the model. The model overestimates the ozone by ~25% across all the locations where
ozone observations were available (Figure 3). This would cause a ~20% larger flux of HOI and
$I_2$ as compared to the observed $O_3$ values. However, reducing the flux by 20% is still not enough
for the model to match the observations. Other uncertainties in the calculation of the inorganic
iodine flux calculation are in the Henry's law of HOI, which has not been measured but is



estimated. Past reports in the Indian Ocean have also questioned the accuracy of the
parameterisation-based sea-to-air flux of iodine species in the Indian Ocean (Inamdar et al.,
2020; Mahajan et al., 2019a). The current model results also suggest that the accuracy of the
flux needs to be revisited, and direct flux observations, which have not been made to date,
would be helpful in quantification of the inorganic iodine emissions.
Additionally, there are other sources of uncertainties which could contribute to the mismatch.
For example, the treatment of the heterogeneous chemistry  has large uncertainties in their
uptake coefficients associated to the ability of the model to simulate the aerosol size
distribution (and aerosol surface area) and the mixing state and surface composition of the
atmospheric aerosols. The photochemistry of $I_2O_x$ species also represents an important source
of uncertainty in the iodine chemical mechanism incorporated into chemistry transport models
(Lewis et al., 2020; Saiz-Lopez and von Glasow, 2012; Sommariva et al., 2012). However, a
new set of $I_xO_y$ photodepletion experiments have recently been reported, but not been
incorporated in the available model mechanisms (Lewis et al., 2020). A further uncertainty on
the IO concentration calculation is that most chemical transport models tend to underestimate
the sources of nitrogen in the open ocean resulting in lower levels of $NO_x$ in the MBL e.g.
Travis et al. (2020), which could lead to higher mixing ratios of IO.
Using a reduced flux, seasonally, the highest levels of IO across the domain are observed during
the winter monsoon period in January, and the lowest levels are observed during the summer
monsoon period in July (Figure 4). While higher values (between 0.7 – 0.9 pptv) are observed
in the Bay of Bengal compared to the Arabian Sea, a clear peak in IO is seen close to 3° N in
the Western Indian Ocean, between 65° E and 70° E. This high is even more prominent during
the pre-monsoon season in April, with the peak monthly averaged values reaching as high as
1.3 pptv. A similar high is also observed during April in the eastern part of the domain close to
the equator. A strong seasonal variation is seen, with IO values significantly lower in July as





compared to January and April. July is the summer monsoon period, and is characterised by
cleaner air over the domain, with clean oceanic air coming from the south-west (Figure 1). This
leads to a reduction in the concentrations of pollutants in the MBL. Considering that the
emission of inorganic iodine is driven by the deposition of $O_3$ at the surface, the reduction in
IO can be attributed to a lower concentration of $O_3$ in the MBL in July (Figure 5). When
averaged over the entire domain, the mean IO mixing ratios are 0.47±0.32 pptv in January,
0.48±0.33 pptv in April and 0.15±0.15 pptv in July, showing the strong seasonality driven by
the emission of inorganic iodine compounds from the ocean surface. When a land mask is
applied and a mean only over the MBL is computed, the values increased to 0.63±0.20 pptv in
January, 0.64±0.22 pptv in April and 0.19±0.14 pptv in July. These values are higher than the
means across the entire domain, showing that most of the IO is restricted to the MBL, close to
the oceanic sources. The concentrations of IO in the current domain are lower than levels
predicted by past studies in other environments. Using a similar setup to the current study in
WRF-Chem, Badia et al. (2019) estimated IO levels of 0.5 pptv in the subtropics as compared
to about 0.8 pptv in the tropics in the MBL. Li et al. (2020) predicted higher levels in the south
China Sea, with IO values ranging between 1 – 3 pptv. By comparison, results from the
Community Multiscale Air Quality Modelling System (CMAQ) predicted peaks of 4-7 pptv in
the coastal regions around Europe, while the open ocean concentrations were below 1 pptv (Li
et al., 2019). Thus, in comparison, the values in the Indian Ocean are lower, especially in July,
than other regions studied hitherto using regional models, implying a reduced impact of iodine
chemistry on the atmosphere in the northern Indian Ocean environment.
The bottom panels in Figure 4 show the difference in IO between the HAL and orgI scenarios.
During January and April, the differences are large, with most of the IO being contributed by
the inorganic emissions. The largest differences (~1.2 pptv) are observed in locations where a
peak is seen in the HAL scenario, closer to the equator. During July, the differences are smaller,



with most of the open ocean MBL showing a smaller increase compared to the other seasons
when the inorganic flux is included. It should however be remembered that even though the
differences in July are only as high as 0.5 pptv, the orgI scenario predicts only up to 0.04 pptv
during this season, which is lower by an order of magnitude. Seasonally also, the difference
between the two scenarios is large, with the domain averaged IO mixing ratios showing values
of 0.46±0.31 pptv in January, 0.47±0.32 pptv in April and 0.14±0.14 pptv in July for the HAL-
orgI contribution. When a land mask is applied and a mean only over the MBL is computed,
the values are 0.62±0.20 pptv in January, 0.63±0.21 pptv in April and 0.18±0.14 pptv in July.
This suggests that most of the IO in the Indian Ocean MBL is due to emissions of inorganic
iodine compounds, rather than the photolysis of organoiodides, which are long-lived and hence
do not contribute heavily to the MBL. A similar result has been observed in other oceanic
MBLs, where observations show that a small fraction of the total IO in the MBL is due to
organic compounds  (Mahajan et al., 2010b).
For the rest of the analysis, we use the difference between the HAL and BASE scenarios to
quantify the impact of iodine chemistry considering that the orgI greatly underestimates the
iodine concentrations in the model domain. The differences and percentage differences in
oxidising species such as ozone ($O_3$), nitrogen oxides ($NO_2$ and NO), hydrogen oxides (OH
and $HO_2$) and the nitrate radical ($NO_3$) are studied to quantify the impact of iodine on the MBL
atmosphere.
**3.3 Impact on ozone**
Figure 5 shows the geographical distribution of $O_3$ across the selected domain during the three
seasons for the HAL scenario, along with the absolute and percentage difference between the
HAL and BASE scenarios. As expected, much higher concentrations of $O_3$ are observed over
the Indian subcontinent as compared to in the surrounding ocean MBL, with $O_3$ mixing ratios





peaking over 50 ppbv in the subcontinent as compared to mixing ratios less than 10 ppbv in
certain parts of the MBL. A steady decrease is observed from the coast to the open ocean
environment, which is expected considering that the main sources of $O_3$ are emitted on the
subcontinent. Seasonal changes are observed, with higher concentrations observed during
January and April as compared to the summer monsoon period. During January and April, the
winds flow from the subcontinent towards the open ocean, while during July the winds flow
from the open ocean towards the subcontinent, which results in cleaner air masses during July
(Figure 1). Additionally, during the summer monsoon, wet deposition also plays a role in the
removal of $O_3$ and its precursors from the atmosphere. During January, higher values are
observed in the MBL, which is due to stronger winds advecting the polluted air masses from
the continent (Figure 1). The model also predicts higher values of $O_3$ over the Bay of Bengal
as compared to the Arabian Sea, which was also seen in the observations (Figure 2). When
averaged over the entire domain, the mean $O_3$ mixing ratios are $32.16\pm9.76$ ppbv in January,
$29.64\pm10.79$ ppbv in April, and $23.34\pm8.85$ ppbv in July for the HAL scenario. The lowest
values are seen during the monsoon period when cleaner oceanic air is seen over the domain
(Table 2). If only the MBL, where elevated concentrations of IO are observed, is considered,
the mean $O_3$ mixing ratios are $28.17\pm7.83$ ppbv in January, $24.17\pm6.42$ ppbv in April, and
$19.49\pm5.97$ ppbv in July. This shows that the advection of anthropogenic sources from the
continent affects the ozone in the remote MBL too (Table 2).
The middle panels of Figure 5 show the absolute difference in $O_3$ over the model domain.
During January and April significant ozone destruction is observed in the MBL, with as much
as 3.5 ppbv lower $O_3$ in the HAL scenario as compared to the BASE scenario. During January
relatively larger destruction is observed in the Bay of Bengal as compared to the Arabian Sea.
Significant losses in $O_3$ are also observed in the western Indian Ocean closer to the equator.
Interestingly, $O_3$ destruction is also visible over the Indian subcontinent, showing that the





effects of iodine chemistry are not just limited to the MBL. In January, an increase of 1 ppbv
is seen in over the south of India. During April, the destruction of $O_3$ is more restricted to the
MBL, with larger destruction observed in the Arabian Sea as compared to the Bay of Bengal.
The main difference in the $O_3$ during these two months is driven by the dynamics which dictates
where the oceanic emissions of iodine are advected. During July, negligible difference is
observed between the HAL and BASE case, with the depletion within ±0.3 ppbv, which reflects
the lower concentrations of iodine during the summer monsoon period. When averaged over
the entire domain, the mean change in $O_3$ mixing ratios are -1.20±0.77 ppbv in January, -
0.97±0.71 ppbv in April and a small increase of 0.01±0.31 ppbv in July, showing that through
January and April a mean reduction is observed, but in July there is a statistically non-
significant increase when the IO concentrations are very low. This change in ozone is mainly
driven by changes in the MBL, where the differences are -1.31±0.67 ppbv in January, -
1.22±0.65 ppbv in April and -0.10±0.21 ppbv in July. If we consider the absolute changes
rather than the mean change, the average over the whole domain is larger at 1.25±0.69 ppbv in
January, 0.98±0.69 ppbv in April and 0.21±0.22 ppbv in July, while over the MBL the
differences are 1.31±0.66 ppbv in January, 1.22±0.65 ppbv in April and 0.15±0.18 ppbv in July
(Table 3). The reason for larger absolute differences as compared to mean differences is that
there are both increases and decreases seen through the domain, and hence the absolute
differences gives us an idea of the total impact of iodine chemistry.
The bottom panels in Figure 5 show the percentage change in $O_3$ between the BASE and HAL
scenarios. As much as 20% reduction in the $O_3$ concentrations is observed in the MBL when
iodine chemistry is included, with the largest differences observed in the western part of the
domain, closer to the equator. For most of the domain, the change in $O_3$ is <15%. Over the
Indian subcontinent, and close to the coastal areas, the relative change in $O_3$ is small, which is
due to larger absolute concentrations in these locations. In January and July, a small increase



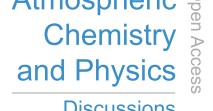

(<5%) in the $O_3$ concentrations is predicted over large parts of the domain. This shows the non-
linear effect of iodine chemistry on the atmosphere, which can lead to an increase in $O_3$ in
certain parts of the domain due to changes in other oxidants. When averaged over the entire
domain, the mean percentage changes in $O_3$ mixing ratios are -3.60±3.33 % in January, -
3.16±3.62 % in April and 0.06±1.37 % in July, showing that the largest relative change is seen
during the winter period in January followed by the pre-monsoon season in April, while the
smallest change is in July during the monsoon when the IO values are low (Table 2). Over the
MBL, the mean percentage changes in $O_3$ mixing ratios are -4.43±3.39 % in January, -
4.80±3.49 % in April, and -0.51±1.26 in July, which means that larger differences are seen in
the MBL rather than over the Indian subcontinent (Table 2). When the absolute change is
computed, the differences are similar with the average over the whole domain showing a
3.75±3.17 % change in January, 3.21±3.58 % change in April and 0.89±1.04 % change in July,
and only over the MBL, the absolute percentage change is 4.45±3.37 % in January, 4.80±3.49
% in April and 0.77±1.13 % in July (Table 3). The fact that the absolute change values are
close to the mean change values shows that most of the domain sees a destruction in ozone due
to the presence of iodine compounds.
This relative change is lower than in the Pacific, where the WRF-Chem simulated $O_3$
destruction because of all halogens peaked at -16 ppbv, which was approximately 70% of the
total ozone loss, of which 18-23% was because of iodine chemistry (Badia et al., 2019). The
loss of $O_3$ due to iodine was similar to the current domain in China, where the range of ozone
destruction/production because of all halogens was -10 to +5% (Li et al., 2020). Over Europe,
combined halogen chemistry, which includes I, Br and Cl, significantly reduces the
concentrations of and $O_3$ by as much as 10 ppbv. The contribution because of only iodine is
also larger than in the current domain, which is expected because of the higher IO
concentrations simulated in Europe (Li et al., 2019).





### 3.4 Impact on nitrogen oxides (NOx)

Halogen oxides interact with nitrogen oxides to change the $NO/NO_2$ ratio by reacting with NO to form $NO_2$. Additionally, iodine oxides can also react with $NO_x$ to form iodine nitrate ($IONO_2$), which can be taken up on aerosol surfaces to act as a sink or recycle both nitrogen compounds and iodine compounds (Atkinson et al., 2007). Thus, the resultant increase or decrease in nitrogen oxides depends on the concentrations of iodine compounds, concentrations of nitrogen compounds and the aerosol surface available for heterogenous recycling. Figures 6 and 7 show the geographical distribution of $NO_2$ and NO across the selected domain during the three seasons, for the HAL scenario, along with the absolute and percentage difference between the HAL and BASE scenarios. As expected, much higher concentrations of $NO_2$ are observed over the Indian subcontinent as compared to the surrounding ocean MBL, with $NO_2$ mixing ratios peaking over 5 ppbv in the subcontinent as compared to mixing ratios less than 0.2 ppbv in certain parts of the MBL (Figure 6). The hotspots for $NO_2$, which are either power plants or large cities, are clearly visible. A sharp decrease is observed from the coast to the open ocean environment, which is expected considering that the primary sources of $NO_2$ are on the subcontinent. The shipping lanes in the Indian Ocean also show higher concentrations of $NO_2$, and are clearly visible, especially south of the Indian subcontinent, where $NO_2$ mixing ratios of up to 1 ppbv can be seen. A seasonal variation is also observed, with higher concentrations observed during winter in January, followed by the pre-monsoon period in April, with the summer monsoon period in July showing the lowest concentrations, even at the hotspots. When averaged over the entire domain, the mean $NO_2$ mixing ratios are 0.43±1.27 ppbv in January, 0.30±0.77 ppbv in April, and 0.27±0.79 in July for the HAL scenario, with the lowest values observed during the monsoon period when cleaner oceanic air is seen over the domain (Table 2). Over only the MBL, the mean $NO_2$ mixing ratios are 0.10±0.46 ppbv in January, 0.06±0.30


ppbv in April, and 0.07±0.29 ppbv in July. This shows that the MBL is much cleaner than the
air above the Indian subcontinent (Table 2).
Similar to $NO_2$, NO also shows higher concentrations over the Indian subcontinent as compared
to the surrounding ocean MBL, with NO mixing ratios peaking over 400 pptv in the
subcontinent as compared to mixing ratios less than 20 pptv in large parts of the MBL (Figure
7). The hotspots for NO, which coincide with the hotspots for $NO_2$, are also clearly visible. A
sharp decrease is observed from the coast to the open ocean environment like $NO_2$, indicating
that fossil fuel combustion over land is the main source. The shipping lanes in the Indian Ocean
are more noticeable than for $NO_2$, with NO mixing ratios of up to 200 pptv observed in some
regions. The seasonal variation for NO follows the same trend as $NO_2$, with higher
concentrations observed during January, followed by April, with the summer monsoon period
in July showing the lowest concentrations. However, January shows the lowest concentrations
in the shipping lanes. When averaged over the entire domain, the mean NO mixing ratios are
49.50±221.23 pptv, 36.66±164.95 pptv and 38.79±173.78 pptv in January, April, and July,
respectively (Table 2). The large standard deviations show that the high concentrations of NO
are mainly restricted to hotspots. Over only the MBL, the mean $NO_2$ mixing ratios are
12.56±85.76 pptv in January, 10.38±77.48 pptv in April, and 11.64±58.45 pptv in July. The
lower values and the smaller standard deviations show that the MBL is much cleaner than the
air above the Indian subcontinent and does not contain large hotspots although it is affected by
the coastal regions (Table 2).
The middle panels of Figures 6 and 7 show the absolute difference in $NO_2$ and NO for the HAL
and BASE scenarios. For $NO_2$, a small reduction is observed in most of the MBL during all the
months, with changes of about -0.04 ppbv observed at most locations. Over the subcontinent,
there is variation observed at some locations, with decreases and increases showing a maximum
of ±0.08 ppbv. Over the shipping lanes, where high $NO_2$ is observed, an increase of about 0.04



ppbv is observed after the inclusion of iodine chemistry. In the case of NO, the variation
observed is similar to $NO_2$, with a small reduction observed in most of the MBL during all the
months, with changes of about -2 pptv observed at most locations. Over the subcontinent,
significant variation is also observed for NO, with decreases and increases showing a maximum
of $\pm 8$ pptv. In most of the shipping lanes, where high NO is observed, the inclusion of iodine
chemistry leads to an increase in the $NO_x$ concentrations, especially in April, where the increase
in $NO_2$ can be as high as ~10% and the increase in NO can be as high as 15%. When averaged
over the whole domain, the mean change in $NO_2$ mixing ratios is negligible at $-0.0040\pm0.0209$
ppbv in January, $0.0007\pm0.0195$ ppbv in April and $0.0003\pm0.0129$ ppbv in July. Over the MBL
too, the differences are just $-0.0025\pm0.0071$ ppbv in January, $-0.0005\pm0.0070$ ppbv in April
and $-0.0008\pm0.0061$ ppbv in July (Table 2). This is because $NO_2$ shows an increase in some
regions within the domain and a decrease in other parts. In the case of NO, the average
difference over the whole domain is also small at $-0.23\pm5.83$ pptv in January, $0.59\pm6.49$ pptv
in April and $-0.09\pm4.44$ pptv in July. Over the MBL too, the differences are just $-0.25\pm2.85$
pptv in January, $0.16\pm2.87$ pptv in April and $-0.20\pm1.99$ in July (Table 2). However, similar to
$NO_2$, these values are misrepresentative of the effect of IO because of differences in the sign
of the change across the domain. If we consider the mean absolute difference instead of just
the mean difference, the larger impact of iodine chemistry can be discerned, with a mean
absolute difference in the $NO_2$ mixing ratios of $0.008\pm0.019$ ppbv in January, $0.007\pm0.018$
ppbv in April and $0.004\pm0.012$ ppbv in July across the whole domain and $0.004\pm0.007$ ppbv
in January, $0.002\pm0.007$ ppbv in April and $0.002\pm0.006$ ppbv in July only over the MBL. For
NO, the change is also higher at $1.60\pm5.61$ pptv in January, $1.57\pm6.33$pptv in April and
$1.03\pm4.32$ pptv in July across the whole domain and $0.70\pm2.78$ pptv, $0.72\pm2.78$ pptv and
$0.45\pm1.95$ pptv in only the MBL during January, April and July (Table 3).





The bottom panels in Figures 6 and 7 show the percentage changes in $NO_2$ and NO between
the BASE and HAL scenarios. Significant differences are observed over the MBL with
decreases in $NO_x$ as high as 50% over large areas when iodine chemistry is included. The
largest differences are observed in the western Arabian Sea and in the southern Bay of Bengal.
Over the Indian subcontinent, and close to the coastal areas, the relative change in both $NO_2$
and NO is small, due to larger absolute concentrations in these locations, although a small
increase is observed over most of the land area. In the shipping lanes, $NO_x$ mostly shows an
increase, which is due to the recycling of halogen nitrates on the aerosol surfaces. When
averaged over the whole domain, the mean percentage change in $NO_2$ mixing ratios is small at
-0.91±11.08 %, 0.22±6.89 % and 0.1±5.85 % in January, April, and July. Over the MBL, mean
values of the differences are slightly higher with values of -2.42±11.62 %, -0.91±7.19 % and -
1.19±6.24 % in January, April, and July. This suggests that the inclusion of iodine chemistry
leads to the reduction in $NO_2$ in the domain, albeit with a large variation, which would
contribute to the reduction in $O_3$ as mentioned above since $NO_2$ is the main source of ozone in
the MBL. The change in NO is also small at -0.47±15.32 % in January, 1.64±10.85 % in April
and -0.23±7.35 % in July, with slightly higher values observed when averaged only over the
MBL: 1.96±15.6 % in January, 1.54±11.60 % in April and -1.71±7.73 in July (Table 2). When
we consider the mean absolute change to see the actual impact of iodine chemistry, the values
of the means are much higher, with as much as ~3.5% change in $NO_2$ and 7% change in NO
observed over the MBL (Table 3). This change in $NO_x$ is smaller than simulated in Europe with
$NO_2$ predicted to increase across most of Europe with most regions showing an increase
between 50 – 200 pptv. However, this was the increase reported due to the inclusion of all the
halogens, and the impact of only iodine would be lower, even though higher levels were
simulated across Europe (Li et al., 2019).



### 3.5 Impact on hydrogen oxides ($HO_x$)

Hydrogen oxides are impacted by iodine chemistry through the catalytic reaction involving IO changing $HO_2$ into OH. This leads to an increase in the oxidizing capacity of the atmosphere due to an increase in the OH concentrations. Figures 8 and 9 show the geographical distribution of OH and $HO_2$ across the selected domain during the three seasons for the HAL scenario, along with the absolute and percentage differences between the HAL and BASE scenarios. The daily averaged OH mixing ratios peak at about 0.5 pptv in the MBL close to the Indian subcontinent, as compared to mixing ratios less than 0.3 pptv over most of the subcontinent (Figure 8). The shipping lanes in the Indian Ocean show higher concentrations of OH, and are clearly visible, especially south of the Indian subcontinent and in the Arabian Sea, where OH mixing ratios of up to 0.45 pptv can be seen. A strong seasonal variation is observed as expected, with higher concentrations observed during the months of April and July, with the winter period in January showing the lowest concentrations. This annual variation is driven by the availability of solar radiation, which is a critical component in OH production. When averaged over the entire domain, the mean OH mixing ratios are $0.14\pm0.05$ pptv, $0.26\pm0.07$ pptv and $0.28\pm0.08$ pptv in January, April, and July, respectively (Table 2). Over only the MBL, the mean OH mixing ratios are $0.15\pm0.05$ pptv in January, $0.27\pm0.08$ pptv in April and $0.27\pm0.08$ pptv in July (Table 2).

$HO_2$ shows much higher concentrations over the Indian subcontinent as compared to the surrounding ocean MBL, with $HO_2$ mixing ratios peaking over 15 pptv in the subcontinent as compared to mixing ratios less than 10 pptv over most of the MBL (Figure 9). There is a correlation between the hotspots for $NO_x$, and low concentrations of $HO_2$ over the Indian subcontinent. This is due to the titration of $HO_2$ by NO, which forms $NO_2$ and leads to an increase in $O_3$ formation. A gradual decrease in the $HO_2$ mixing ratios is observed from the subcontinent to the open ocean environment during the months of April and July, although the



$HO_2$ concentrations in the MBL are larger during January. Relatively, the winter month of
January shows the lowest $HO_2$ mixing ratios of the three months. The shipping lanes in the
Indian Ocean are clearly visible like for OH, although the $HO_2$ concentrations in the shipping
lanes are lower than the surrounding areas. This is due to the earlier mentioned titration of $HO_2$
by ship emitted NO, which leads to an increase in OH but a decrease in $HO_2$. When averaged
over the entire domain, the mean $HO_2$ mixing ratios are 7.10±1.49 pptv, 10.18±1.64 pptv and
9.24±1.97 pptv in January, April, and July, respectively (Table 2). Over only the MBL, the
mean $HO_2$ mixing ratios are higher at 7.32±1.12 pptv, in January but lower in April and July
at 9.80±1.36 and 8.67±1.53 pptv (Table 2).
The middle panels of Figures 8 and 9 show the absolute difference in OH and $HO_2$. For OH, a
small increase in the OH concentration is observed in most of the MBL during the months of
January and April, with the largest increase of about 0.03 pptv observed in the Arabian Sea.
However, for most of the domain, the increase in OH is small, with differences of 0.01 pptv
compared to the BASE scenario. During the monsoon month of July, a small decrease is
observed over most of the domain with an increase observed further south close to the equator.
Over the shipping lanes, a small reduction is observed during all the months, with changes of
about -0.02 pptv along the ship tracks. In the case of $HO_2$, a clear land ocean contrast is
observed in the differences, with the $HO_2$ values reducing over the entire MBL, but showing a
small increase over the subcontinent. The largest reduction is observed in the south-western
Arabian Sea, with changes of about -1.8 pptv in the HAL scenario as compared to the BASE
case. Seasonally, the largest changes in $HO_2$ are observed in January, with the least changes
observed in the monsoon month of July. IO concentrations are the lowest during monsoon due
to clean air-masses reducing the ozone deposition driven emissions and hence the difference
between the HAL and BASE scenarios is also the lowest during July. When averaged over the
whole domain, the mean change in OH mixing ratios is negligible at 0.001±0.006 pptv in





January, 0.006±0.007 pptv in April and -0.003±0.006 in July. In the case of $HO_2$, the average
difference over the whole domain is also small at -0.48±0.43 pptv in January, -0.35±0.38 pptv
in April and -0.19±0.16 in July. Over the MBL too, the differences are larger with the largest
difference being -0.67±0.36 pptv in January (Table 2).
The bottom panels in Figures 8 and 9 show the percentage changes in OH and $HO_2$ between
the BASE and HAL scenarios. Significant differences are observed with an increase in OH and
a decrease in the $HO_2$ over most of the MBL. The largest change in OH is observed in the
northern Arabian Sea MBL, with a difference of more than 15% between the HAL and BASE
cases when iodine chemistry is included. Large parts of the Arabian Sea and the Bay of Bengal
show an increase in OH of up to 10% for January and April, with a smaller difference observed
in July due to lower concentrations of iodine compounds in the atmosphere. In January and
April, when the concentrations of IO are higher, a negative change in the OH concentrations
are observed over the shipping lanes. In the case of $HO_2$, a large change of up to 25% is
observed in the MBL, with the largest differences observed in the southern western Arabian
Sea, close to the equator. During the months of January and April, most of the MBL shows a
change of -10 to -20 %, while a positive change of 0-5% is observed over the Indian
subcontinent. The mean percentage change in the OH and $HO_2$ mixing ratios peaks at 2.6 %
and 8.4 % for the months of April and July, respectively (Table 2), but the absolute percentage
change in OH is higher at 3.6 % in January, while the $HO_2$ absolute percentage change (Table
3) is about ~8.4 % showing the large impact of iodine chemistry on the oxidation capacity of
the MBL. For example, the 3.29% increase in the OH concentrations observed across the
domain in January would result in the lowering of the methane lifetime by 3.19% in the MBL
(assuming $k_{CH4+OH} = 1.85×10^{-12}exp(-1690/T)$; (Atkinson et al., 2006)). A similar change in the
oxidizing capacity has been simulated in other parts of the world, with halogen chemistry
inducing complex effects on OH (ranging from −0.023 to 0.030 pptv) and $HO_2$ (in the range of



−3.7 to 0.73 pptv) in Europe (Li et al., 2019) and enhancing the total atmospheric oxidation
capacity in polluted areas of China, typically 10% to 20% (up to 87% in winter) and mainly by
significantly increasing OH levels (Li et al., 2020). The moderate increase in the oxidation
capacity over the northern Indian Ocean and the Indian subcontinent is due to the lower
concentrations of IO in the domain, along with the fact that this number is calculated only for
the impact of iodine chemistry, while the past studies have reported the impact of all halogens.
Globally the average increase in OH because of the inclusion of iodine chemistry has been
estimated to be 1.8 %, which is comparable to the current domain (Sherwen et al., 2016).

**3.6 Impact on the nitrate radical (NO₃)**
$NO_3$ radicals are the predominant night-time oxidant and play a similar role to OH during the
daytime in the degradation of atmospheric constituents (Wayne et al., 1991). Iodine compounds
interact with $NO_3$, mainly through the primary emissions of inorganic iodine compounds by
the oxidation of chemicals such as $I_2$ and HOI (Saiz-Lopez et al., 2016). Figure 10 shows the
geographical distribution of $NO_3$ across the selected domain during the three seasons, for the
HAL scenario, along with the absolute and percentage difference between the HAL and BASE
scenarios. As expected, much higher concentrations of $NO_3$ are observed over the Indian
subcontinent as compared to the surrounding ocean MBL, with $NO_3$ mixing ratios peaking over
40 pptv in the subcontinent as compared to mixing ratios less than 5 pptv in the MBL
surrounding the Indian subcontinent. A sharp decrease is observed from the coast to the open
ocean environment, which is expected considering that the main sources of $NO_3$ are on the
subcontinent and $NO_3$ has a short lifetime due to its high reactivity. The seasonal variation is
the same as $O_3$, with peak values observed over the Indian subcontinent over the month of
April, followed by January. The monsoon month of July displays the lowest concentrations,





due to efficient removal of $NO_x$ and $O_3$ due to wet deposition. Elevated values up to 15 pptv
are also observed along the shipping lanes due to the conversion of ship emitted $NO_x$ into $NO_3$
during the night-time. When averaged over the entire domain, the mean $NO_3$ mixing ratios are
$7.64\pm8.08$ pptv in January, $10.38\pm15.53$ pptv in April and $4.52\pm6.14$ pptv in July for the HAL
scenario. The lowest values are observed during the monsoon period similar to $O_3$ when cleaner
oceanic air is observed over the domain (Table 2). If only the MBL, where lower concentrations
of $NO_x$ are observed is considered, the mean $NO_3$ mixing ratios are $4.47\pm5.44$ pptv in January,
$2.99\pm4.09$ pptv in April, and $2.38\pm3.94$ pptv in July (Table 2).
The middle panels of Figure 10 show the absolute difference in $NO_3$ over the model domain.
During the months of January and April, a significant reduction of up to -1.5 pptv is observed
in the MBL. During January, a reduction is observed in the Bay of Bengal as well as the Arabian
Sea, but in April the reduction in $NO_3$ is largely observed in the Arabian Sea. This correlates
well with the IO distribution which also shows more iodine activity in the Arabian Sea during
April. A reduction in $NO_3$ is also visible over the Indian subcontinent, and like $O_3$ show that
the effects of iodine chemistry are not just limited to the MBL. Indeed, there are pockets of an
increase in $NO_3$ observed over the subcontinent. During July, negligible difference is observed
between the HAL and BASE case, with a smaller than 0.5 pptv decrease seen across the MBL.
However, during the same period, an increase of up to 1.5 pptv can be seen over the $NO_x$
hotspots over the Indian subcontinent. Decreases of up to -1.5 pptv are also observed along the
shipping lanes, showing the strong interaction between iodine and $NO_x$ chemistry. Over the
whole domain, the inclusion of iodine chemistry results in a mean decrease of about ~-0.4 pptv,
which is slightly higher when a mean is taken only for the MBL (Table 2). The absolute change
in $NO_3$ is even higher, with $NO_3$ values changing by an average of 0.5 pptv across the whole
domain in July (Table 3). This value is however lower than the effect of all the halogens, as





shown by Li et al. (2019) in Europe, where halogens significantly reduced the concentrations
of NO$_3$ by as much as 20 pptv.
The bottom panels in Figure 10 show the percentage change in NO$_3$ between the BASE and
HAL scenarios. As much as a 50% reduction in the NO$_3$ concentrations is observed in the MBL
when iodine chemistry is included, with the largest differences observed in the Arabian Sea,
close to the Indian subcontinent, further west closer to the equator and in the Bay of Bengal.
For most of the other domain, the change in NO$_3$ is <20%. Over the Indian subcontinent, the
relative change in NO$_3$ is small, due to larger absolute concentrations and in some places a
small increase (<5%) is predicted, especially in July when iodine chemistry is not highly active.
The relative change in the shipping lanes is smaller than the surrounding areas due to the higher
relative concentrations of NO$_3$ along the tracks (<20%). On average, the inclusion of iodine
chemistry can cause an almost 10% change in the NO$_3$ concentrations across the MBL in
January with smaller changes of ~4.5% observed during July when the IO concentrations are
lower (Table 3).

**4. Conclusions**
In this study, we used the WRF-Chem regional model to quantify the impacts of the observed
levels of iodine on the chemical composition of the MBL over the northern Indian Ocean. The
model results show that the IO concentrations are greatly underestimated if only organic iodine
compound emissions are considered in the model. This reaffirms that emissions of inorganic
species resulting from the deposition of ozone on the sea surface are needed to reproduce the
observed levels of IO. However, the current parameterisations overestimate the concentrations,
which could be because of a combination of modelling uncertainties and the HOI and I$_2$ flux
parameterisation not being directly applicable to this region. This agrees with previous reports





in the Indian Ocean questioning the current parameterisations and highlights the need for direct
HOI and $I_2$ flux observations. For a reasonable match with cruise-based observations, the
inorganic emissions had to be reduced by 40%. The simulations after this reduction in flux
show a strong seasonal variation, with lower iodine concentrations predicted when cleaner air
is found over the Indian subcontinent due to flushing by remote oceanic air masses during the
monsoon season but higher iodine concentrations are predicted during the winter period, when
polluted air from the Indian subcontinent increases the ozone concentrations in the MBL. A
large regional variation is observed in the IO distribution, and also in the impacts of iodine
chemistry. Iodine catalysed reactions can lead to significant regional changes with peaks of
25% destruction in $O_3$, altering the NOx concentrations by up to 50%, increasing the OH
concentration by as much as 15%,  reducing the $HO_2$ concentration by as much as 25%, and up
to a 50% change in the nitrate radical ($NO_3$). When averaged across the whole domain, the
differences are smaller, although still significant. For example, the average change in OH
across the whole domain reduces the methane lifetime by ~3% in the MBL showing the impact
of iodine on the oxidation capacity. Most of the large relative changes are observed in the open
ocean MBL but iodine chemistry also affects the chemical composition in the coastal
environment and over the Indian subcontinent. Indeed, in some instances an increase in $O_3$
concentrations is predicted over the subcontinent, showing the non-linear effects. These model
results highlight the importance of iodine chemistry in the northern Indian Ocean and suggest
that it needs to be included in future studies for improved accuracy in modelling the chemical
composition in this region.

**Acknowledgements**



IITM is funded by the Ministry of Earth Sciences (MOES), Government of India. This study
has been funded by the European Research Council Executive Agency under the European
Union´s Horizon 2020 Research and Innovation programme (Project 'ERC-2016-COG 726349
CLIMAHAL').

**Author Contributions**

ASM conceptualized the research plan and methodology, analysed the data, and wrote the
paper. QL did the model runs for the study and contributed towards the interpretation of the
results and writing. SI helped with the analysis, interpretation and writing. KR, AB and ASL
contributed towards the interpretation and writing.

**Competing interests**

The authors declare that they have no conflict of interest.

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





**Tables**
**Table 1:** Monthly mean of IO concentration in parts per trillion by volume (pptv) over the
domain region for model simulations in January, April, and July 2015, and simulation scenarios
orgI, HAL and difference between HAL-orgI, before and after applying a land mask over the
model domain.

| IO (pptv) | Jan | April | July |
|---|---|---|---|
| **Over the whole domain** | | | |
| orgI | 0.011±0.009 | 0.008±0.006 | 0.012±0.009 |
| HAL | 0.47±0.32 | 0.48±0.33 | 0.15±0.15 |
| HAL-orgI | 0.46±0.31 | 0.47±0.32 | 0.14±0.14 |
| **Only over the MBL** | | | |
| orgI | 0.015±0.009 | 0.011±0.006 | 0.015±0.008 |
| HAL | 0.63±0.20 | 0.64±0.22 | 0.19±0.14 |
| HAL-orgI | 0.62±0.20 | 0.63±0.21 | 0.18±0.14 |




**Table 2:** Monthly means and standard deviations of $O_3$, $NO_2$, NO, $NO_3$, OH, $HO_2$ mixing
ratios (unit in parenthesis) over the domain region for the model simulations in January,
April, and July for the HAL scenario along with the difference and percentage differences
between HAL and BASE. The table also includes monthly mean values only over the
MBL.

| | January | April | July | January | April | July |
|---|---|---|---|---|---|---|
| | $O_3$ (ppbv) | | | $NO_3$ (pptv) | | |
| | Over the whole domain | | | | | |
| **HAL** | 32.16±9.76 | 29.64±10.79 | 23.34±8.85 | 7.64±8.08 | 10.38±15.53 | 4.52±6.14 |
| **HAL-BASE** | -1.20±0.77 | -0.97±0.71 | 0.01±0.31 | -0.39±0.43 | -0.33±0.83 | -0.03±0.29 |
| **HAL-BASE %** | -3.6±3.33 | -3.16±3.62 | 0.06±1.37 | -4.85±14.07 | -3.09±10.72 | -0.64±8.08 |
| | Only over the MBL | | | | | |
| **HAL** | 28.17±7.83 | 24.17±6.42 | 19.49±5.97 | 4.47±5.44 | 2.99±4.09 | 2.38±3.94 |
| **HAL-BASE** | -1.31±0.67 | -1.22±0.65 | -0.10±0.21 | -0.43±0.34 | -0.27±0.31 | -0.08±0.19 |
| **HAL-BASE %** | -4.43±3.39 | -4.80±3.49 | -0.51±1.26 | -8.80±14.41 | -8.23±10.49 | -3.14±8.29 |
| | $NO_2$ (ppbv) | | | OH (pptv) | | |
| | Over the whole domain | | | | | |
| **HAL** | 0.43±1.27 | 0.30±0.77 | 0.27±0.79 | 0.14±0.05 | 0.26±0.07 | 0.28±0.08 |
| **HAL-BASE** | -0.0040±0.0209 | 0.0007±0.0195 | 0.0003±0.0129 | 0.001±0.006 | 0.006±0.007 | -0.003±0.006 |
| **HAL-BASE %** | -0.91±11.08 | 0.22±6.89 | 0.10±5.85 | 0.34±4.54 | 2.55±2.47 | -0.94±2.22 |
| | Only over the MBL | | | | | |
| **HAL** | 0.10±0.46 | 0.06±0.30 | 0.07±0.29 | 0.15±0.05 | 0.27±0.08 | 0.27±0.08 |
| **HAL-BASE** | -0.0025±0.0071 | -0.0005±0.0070 | -0.0008±0.0061 | 0.001±0.007 | 0.007±0.007 | -0.002±0.006 |
| **HAL-BASE %** | -2.42±11.62 | -0.91±7.19 | -1.19±6.24 | 0.44±5.06 | 2.62±2.35 | -0.67±2.23 |
| | NO (pptv) | | | $HO_2$ (pptv) | | |
| | Over the whole domain | | | | | |
| **HAL** | 49.49±221.23 | 36.66±164.95 | 38.79±173.78 | 7.10±1.49 | 10.18±1.64 | 9.24±1.97 |
| **HAL-BASE** | -0.23±5.83 | 0.59±6.49 | -0.09±4.44 | -0.48±0.43 | -0.35±0.38 | -0.19±0.16 |
| **HAL-BASE %** | -0.47±15.32 | 1.64±10.85 | -0.23±7.35 | -6.39±5.54 | -3.28±4.04 | -2.03±1.71 |
| | Only over the MBL | | | | | |
| **HAL** | 12.56±85.76 | 10.38±77.48 | 11.64±58.45 | 7.32±1.12 | 9.80±1.36 | 8.67±1.53 |
| **HAL-BASE** | -0.25±2.85 | 0.16±2.87 | -0.20±1.99 | -0.67±0.36 | -0.53±0.26 | -0.23±0.14 |
| **HAL-BASE %** | -1.96±15.6 | 1.54±11.6 | -1.71±7.73 | -8.36±4.56 | -5.14±3.05 | -2.60±1.53 |






**Table 3:** Monthly means and standard deviations of $O_3$, $NO_2$, NO, $NO_3$, OH, $HO_2$ mixing
ratios (unit in parenthesis) over the domain region for the model simulations in January,
April, and July for the HAL scenario along with the absolute difference and absolute
difference percentage between HAL and BASE. The table also includes monthly mean
values only over the MBL.

| | January | April | July | January | April | July |
|---|---|---|---|---|---|---|
| | $O_3$ (ppbv) | | | $NO_3$ (pptv) | | |
| | Over the whole domain | | | | | |
| HAL | 32.16±9.76 | 29.64±10.79 | 23.34±8.85 | 7.64±8.08 | 10.38±15.53 | 4.52±6.14 |
| HAL-BASE | 1.25±0.69 | 0.98±0.69 | 0.21±0.22 | 0.46±0.35 | 0.50±0.74 | 0.16±0.25 |
| HAL-BASE % | 3.75±3.17 | 3.21±3.58 | 0.89±1.04 | 5.73±13.6 | 4.68±9.78 | 3.52±6.84 |
| | Only over the MBL | | | | | |
| HAL | 28.17±7.83 | 24.17±6.42 | 19.49±5.97 | 4.47±5.44 | 2.99±4.09 | 2.38±3.94 |
| HAL-BASE | 1.31±0.66 | 1.22±0.65 | 0.15±0.18 | 0.45±0.31 | 0.29±0.29 | 0.11±0.17 |
| HAL-BASE % | 4.45±3.37 | 4.80±3.49 | 0.77±1.13 | 9.20±14.15 | 8.81±10.08 | 4.55±7.47 |
| | $NO_2$ (ppbv) | | | OH (pptv) | | |
| | Over the whole domain | | | | | |
| HAL | 0.43±1.27 | 0.30±0.77 | 0.27±0.79 | 0.14±0.05 | 0.26±0.07 | 0.28±0.08 |
| HAL-BASE | 0.008±0.019 | 0.007±0.018 | 0.004±0.012 | 0.004±0.004 | 0.008±0.005 | 0.005±0.004 |
| HAL-BASE % | 1.89±10.63 | 2.15±5.71 | 1.63±5.33 | 3.29±3.14 | 3.02±1.94 | 1.82±1.40 |
| | Only over the MBL | | | | | |
| HAL | 0.10±0.46 | 0.06±0.30 | 0.07±0.29 | 0.15±0.05 | 0.27±0.08 | 0.27±0.08 |
| HAL-BASE | 0.004±0.007 | 0.002±0.007 | 0.002±0.006 | 0.005±0.004 | 0.008±0.005 | 0.005±0.004 |
| HAL-BASE % | 3.47±11.25 | 3.51±6.08 | 2.52±5.95 | 3.64±3.39 | 3.05±1.89 | 1.75±1.36 |
| | NO (pptv) | | | $HO_2$ (pptv) | | |
| | Over the whole domain | | | | | |
| HAL | 49.49±221.23 | 36.66±164.95 | 38.79±173.78 | 7.10±1.49 | 10.18±1.64 | 9.24±1.97 |
| HAL-BASE | 1.60±5.61 | 1.57±6.33 | 1.03±4.32 | 0.51±0.39 | 0.44±0.27 | 0.21±0.14 |
| HAL-BASE % | 3.22±14.32 | 4.36±8.15 | 2.64±6.46 | 6.76±4.87 | 4.16±3.09 | 2.18±1.52 |
| | Only over the MBL | | | | | |
| HAL | 12.56±85.76 | 10.38±77.48 | 11.64±58.45 | 7.32±1.12 | 9.8±1.36 | 8.67±1.53 |
| HAL-BASE | 0.70±2.78 | 0.72±2.78 | 0.45±1.95 | 0.67±0.36 | 0.53±0.25 | 0.23±0.14 |
| HAL-BASE % | 5.48±14.86 | 7.07±8.58 | 3.76±7.15 | 8.38±4.51 | 5.17±2.98 | 2.63±1.49 |






**Figures**

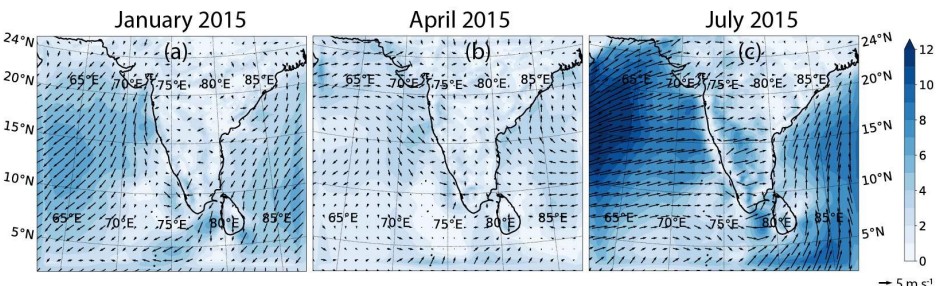

**Figure 1:** The wind direction and speed over the model domain during the three months used to study the impact of iodine chemistry on the marine boundary layer. The three months represent different seasons: the winter monsoon period in January, pre-monsoon in April and the summer monsoon in July. The direction of the arrows shows the wind direction, and the size of the arrows and the contour colours show the magnitude of the wind.






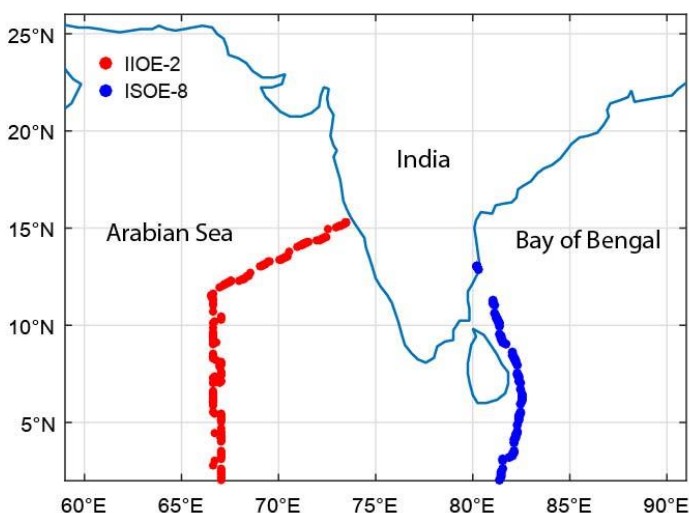

**Figure 2:** The domain chosen for the model runs along with the tracks of the cruises from which data was used for validation are shown. The two cruises were the 2[nd] International Indian Ocean Expedition (IIOE-2; December 2015) and the 8[th] Indian Southern Ocean Expedition (ISOE-8; January 2015) and started from the West and East coast of India, respectively.










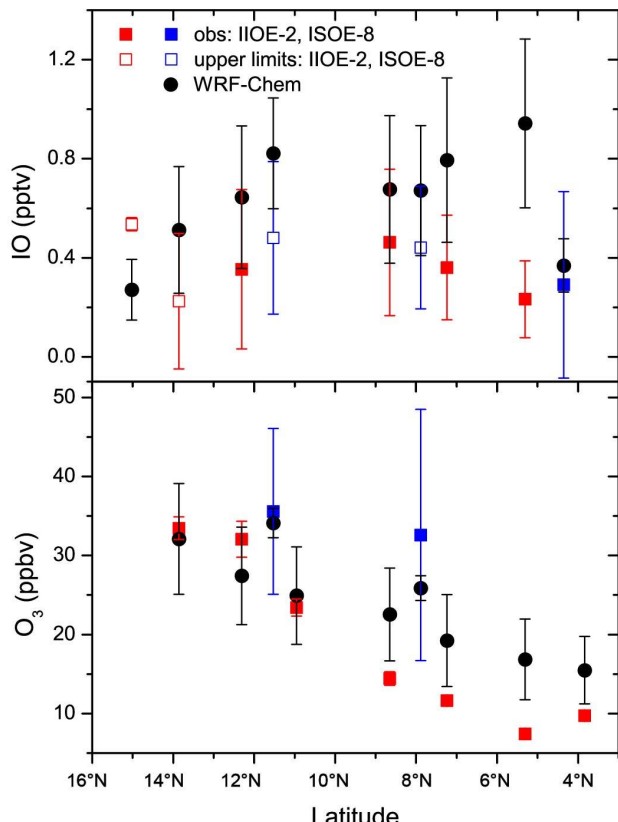


**Figure 3:** A comparison between the model simulated and observed IO (top panel) and O₃
(bottom panel) is shown. In cases where IO was not detected, an upper limit (empty squares)
was estimated.





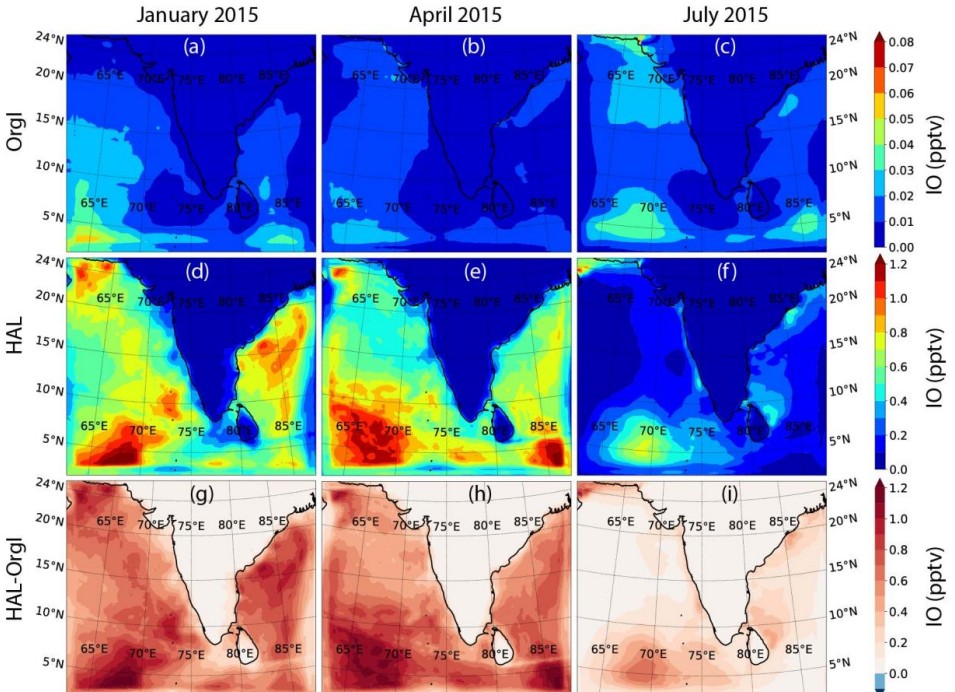

**Figure 4:** Model simulation showing the boundary layer averaged IO mixing ratios across the

domain during the three seasons, along with a difference between the HAL and orgI scenarios

for each season are shown.





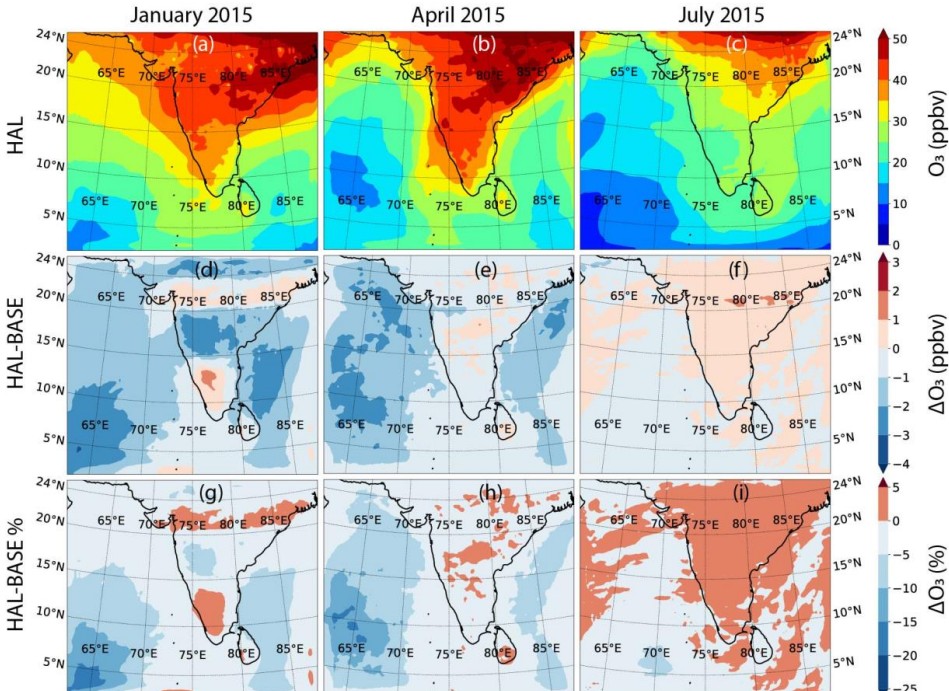

960

**Figure 5:** Model simulations showing the boundary layer averaged O$_3$ mixing ratios across the

domain during the three seasons for the HAL scenario (top panels), along with the differences

(middle panels) and the percentage differences (lower panels) between the HAL and BASE

scenarios for each season are shown.

965





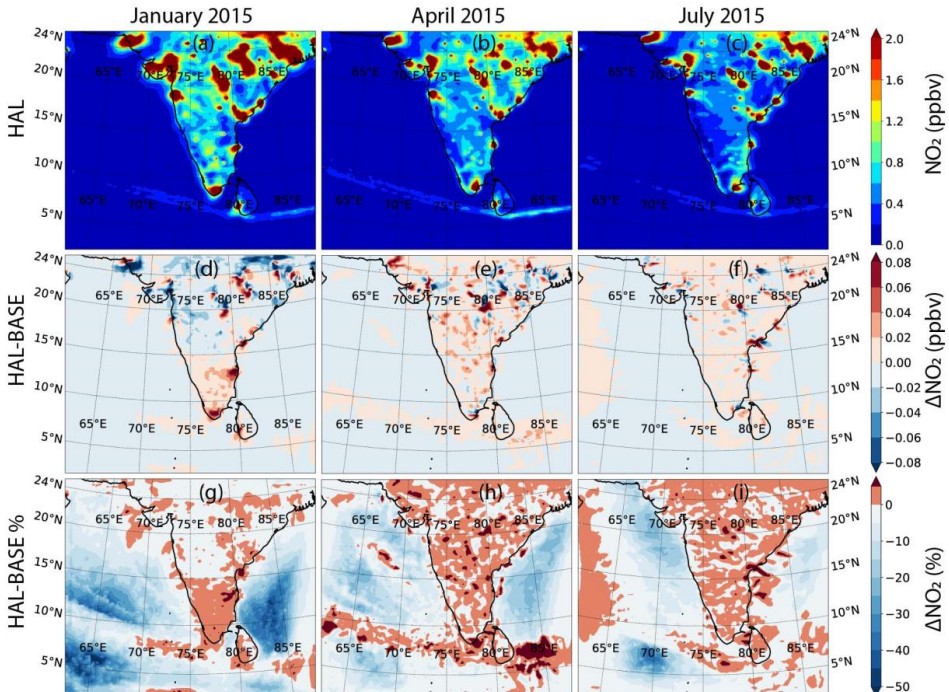

966

**Figure 6:** Model simulations showing the boundary layer averaged NO$_2$ mixing ratios across the domain during the three seasons for the HAL scenario (top panels), along with the differences (middle panels) and the percentage differences (lower panels) between the HAL and BASE scenarios for each season are shown.

971





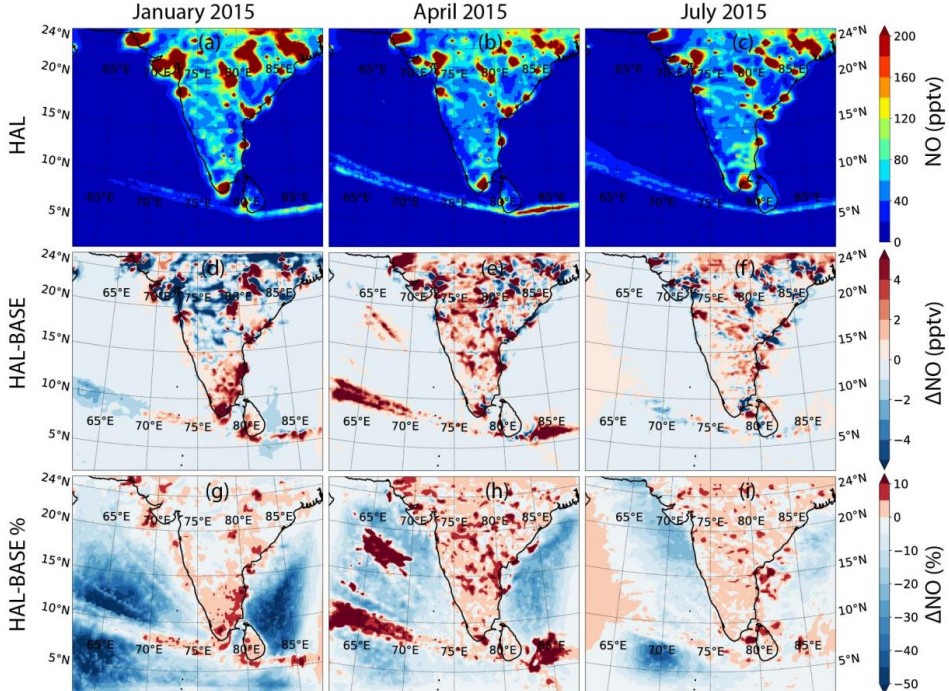

972

**Figure 7:** Model simulations showing the boundary layer averaged NO mixing ratios across the domain during the three seasons for the HAL scenario (top panels), along with the differences (middle panels) and the percentage differences (lower panels) between the HAL and BASE scenarios for each season are shown.

977





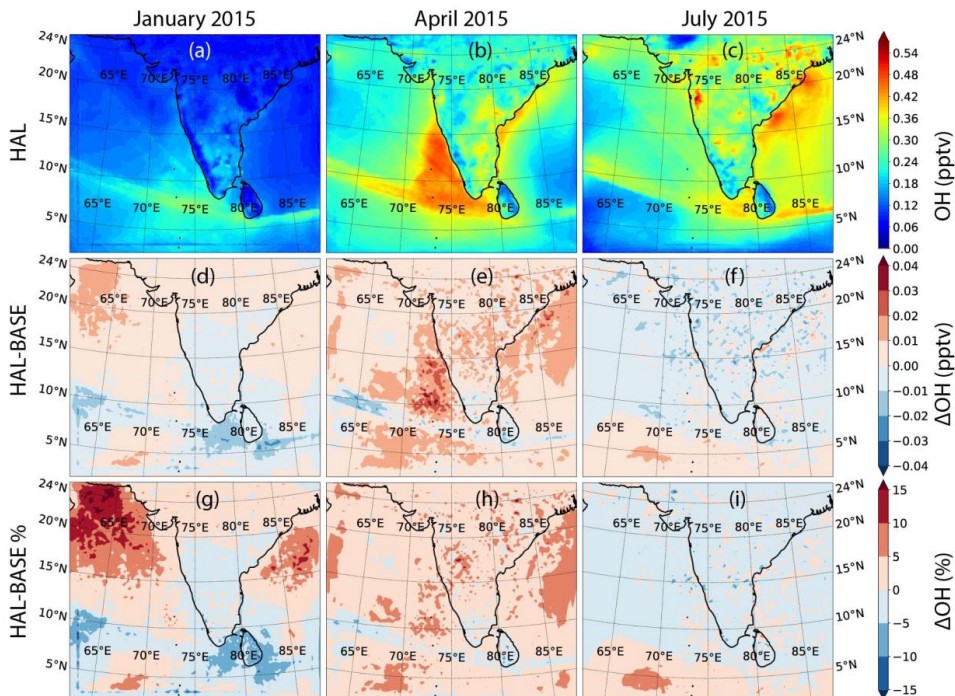

978

**Figure 8:** Model simulations showing the boundary layer averaged OH mixing ratios across the domain during the three seasons for the HAL scenario (top panels), along with the differences (middle panels) and the percentage differences (lower panels) between the HAL and BASE scenarios for each season are shown.

983



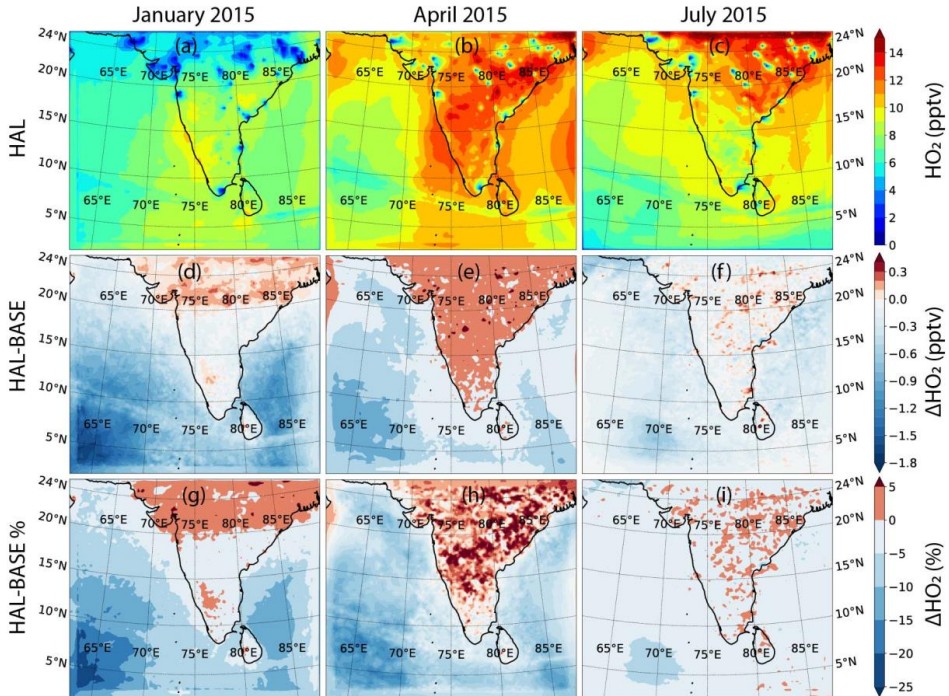

984

**Figure 9:** Model simulations showing the boundary layer averaged $HO_2$ mixing ratios across
the domain during the three seasons for the HAL scenario (top panels), along with the
differences (middle panels) and the percentage differences (lower panels) between the HAL
and BASE scenarios for each season are shown.





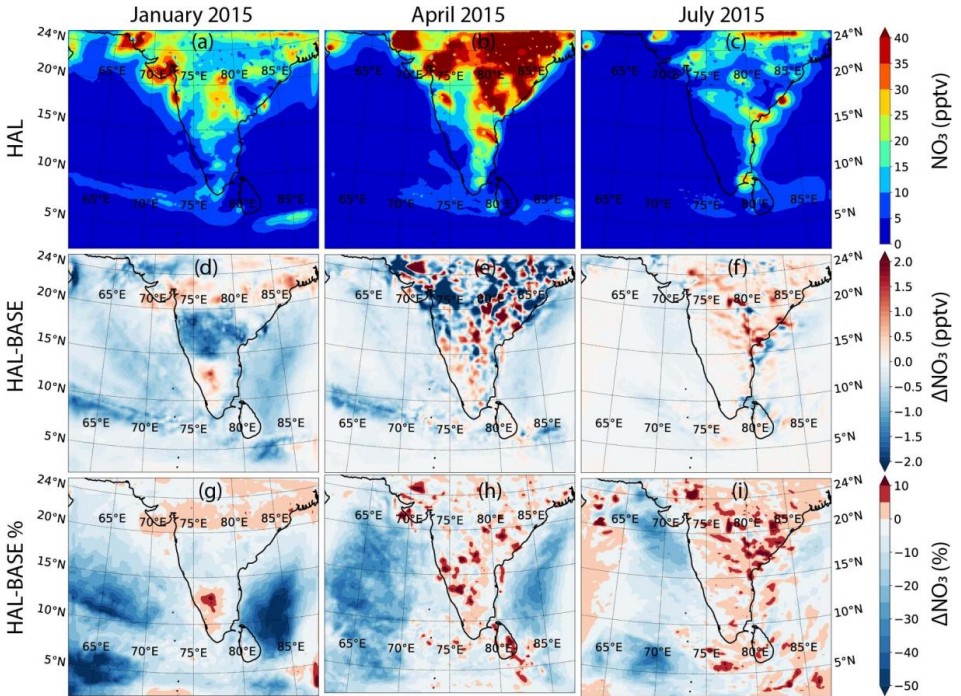

989

**Figure 10:** Model simulations showing the boundary layer averaged $NO_3$ mixing ratios across the domain during the three seasons for the HAL scenario (top panels), along with the differences (middle panels) and the percentage differences (lower panels) between the HAL and BASE scenarios for each season are shown.

994