# Peer review of "Modelling the Impacts of Iodine Chemistry on the Northern Indian Ocean Marine 2 Boundary Layer Anoop S. Mahajan1\*, Qinyi Li2, Swaleha Inamdar1,3, Kirpa Ram3, Alba Badia4 and Alfonso 3 4 Saiz-Lopez2 5 6 1Indian Instit"

_Atmospheric Chemistry and Physics, 2020_

## Referee Comment (RC1) · Rafael Pedro Fernandez (Referee) · 13 Jan 2021

**Review of "Modelling the Impacts of Iodine Chemistry on the Northern Indian Ocean Marine Boundary Layer" by Mahajan et al., ACPD, 2020**

The paper presents a computational evaluation of the impact of oceanic iodine emissions (both organic and inorganic) on the composition of the lower atmosphere, with a special focus on the changes on ozone, HOx and NOx (plus NO3) over the Indian ocean and sub-continent. The most relevant results are: *i)* the requirement to reduce by 40% the strength of the oceanic emissions of inorganic iodine in comparison with previous modeling studies to properly reproduce IO observations performed over the Indian ocean; and ii) the different seasonal impact of iodine chemistry predicted during the pre-monsoon and monsoon periods, mostly due to the changes on the total iodine burden during the different seasons. The paper is well organized and the results are presented in order, although in many sections the writing style, figures format and absolute/percentage change description is repetitive, without a comprehensive interpretation on how most of the modeled changes for each individual species are correlated with the others. This work is certainly of interest and fulfills the requirements and objectives of Atmospheric Chemistry and Physics. However, major changes must be performed before acceptance for publication as described below.

**Major Comments:**

1. Model Validation:
   The paper organization and results presentation is correct, starting with the model validation and followed by a model analysis. However, many technical details are missing (both for measurements data and model output processing), which prevents the reader to understand how well the model compares with observations. For example:
   a. How the cruise observations were averaged? Every 4º Latitdue? Every 2º Latitude? By looking at Figure 2 (ship-tracks) and Figure 3 (IO and O3 vmr), it is evident that IIOE-2 and ISOE-8 observations were post-processed in a different way, but no details are given. Which is the temporal resolution of the observations, and how many days of measurements were considered?
   b. How the model output was extracted and processed to compare with observations? As can be seen in Fig. 3, only 1 dataset of model output is compared with 2 independent cruises. As IIOE-2 (Arabian sea) and ISOE-8 (bay of Bengal) campaigns were performed in different seas (and many of the results presented in Figs. 4-10 highlight differences among these regions), wouldn't it be better to compare model vs. observations for each cruise independently?
   c. Did you extract only the closest model pixels to the location of the cruise or did you average a larger region surrounding the ship-lane and/or the whole oceanic domain? Did you consider the whole monthly mean model output to compare with observations, or did you use only a small time-window equivalent to the one for observations? In case the former option, how is the model variability between the monthly mean and the observations-window time period?

d. The paper presents a model analysis for three different seasons (pre-monsoon in April, summer monsoon in July and winter monsoon in January), while model validation is performed only for winter monsoon when IIOE-2 (December) and ISOE-8 (January) cruises took place. Even when this can be inferred from the text, I suggest explicitly pointing at the lacking of a complete seasonal validation of model results within the model validation section as well as in the Figure caption and discussion. Additionally It would be useful to provide a clear statement within the conclusions that the addressed model impacts for the remaining seasons should be taken with caution, as none the ozone nor the IO results were validated during those seasons.

2. Modeled Geographical Distributions and Statistics:
Once the model is validated, a series of Figures, Tables and Values describing the impact of HAL vs. BASE simulation is provided. Even when in general the presentation is consistent, many of the results are repetitive and presented as a consecutive sequence of absolute and percentage impacts, distinguishing between the whole domain and when a landmask is applied, and written using almost the same style. I believe this description can be improved, making the text more easy-going, pointing at the proper table/figure when results are presented, and mostly by describing absolute/relative impacts altogether as well as making inter-connections between the changes on the abundances of the different species analyzed. In particular:

a. Many paragraphs of sections 3.4 (NOx), 3.5 (HOx) and 3.6 (NO3) start with "The middle panels of Fig. # show the absolute difference …" and/or "The bottom panels in Fig. # shows the percentage change …". Many portions of this sections present a large series of model mean values ± standard deviation that can be found in Tables 2 and 3, and in some cases no additional interpretation to the number values is provided. Even when this style can be helpful at the beginning of the results description to orientate the reader on how Figures and Tables should be read, as the paper moves forward it would be very useful not to repeat the full list of values on the text, but just use those that are necessary to justify the analysis being described.

b. Something similar is present when mean results above the MBL are compared with those over the Indian-subcontinent or the whole model domain. Quantitative results are literally copied from the table into the text for all species and simulations, and usually equivalent conclusions are provided. I found this could be largely reduced to avoid unnecessary repetition, and only include this type of explicit analysis when a special issue must be highlighted, leaving for the remaining of the text a more general comment an inter-connection to the results for other species.

c. In P15/L345-346 and Table 3 (as well as in other sections below), the "absolute changes rather than the mean changes" are presented. The exact difference between each of the magnitudes should be clearly described at first usage. My understanding is that the "mean change" is the signed difference (HAL-BASE), while the "absolute change" is the unsigned module-difference |HAL-BASE|. If that is the case, I do not completely understand the rationale for providing the

unsigned difference, as for most of the results presented here using a signed analysis that allows determining a positive or negative deviation (bias) with respect to a base simulation would be sufficient. If the authors are interested in presenting statistical evaluation of the model performance (comparing HAL sensitivity respect to the BASE simulation), other statistical measures such as the Normalized Fractional Error (NME) or root mean squared error (RMSE) could be used (see Willmott, 1981. On the validation of models. Phys. Geogr. 2, 184–194.https://doi.org/10.1080/02723646.1981.10642213).

d. Model results are provided for the mean volume mixing ratio within the lower 10 layers within the MBL (P7L156) instead of the surface mean. Even when I'm fine with using this procedure as it allows addressing a more realistic indication of the atmospheric impacts of introducing the organic and inorganic iodine sources on the model, it would be great if the authors could provide an indication of how variable is the vertical distribution within the MBL. For example, all values in the Tables and text present the standard deviation of the model results, but those values are more representative of the "spatial" averaging than the "vertical" averaging. Authors should decide whereas it is worth including an additional figure showing vertical profile or latitudinal cross-section across each of the cruises, but at least an indication of the magnitude of the vertical changes within the MBL would be useful.

3. Iodine Oceanic Source vs Iodine wet deposition:
The authors attribute the strong seasonal variation only to the seasonal change in iodine sources driven by the cleaner oceanic air (lower ozone) during summer Monsoon (P11L258-P12/L266). However, they do not provide any quantitative estimation of the overall source of HOI and I2 from the ocean during each season, presenting only changes in the IO vmr within the MBL. Even though large seasonal changes on emissions are expected, the Monsoon drives also large changes in precipitation, which in turn will impact on the washout/wet-scavenging of soluble species (and many iodine species are soluble). The authors should be able to quantify the net change in both the fluxes and also the sinks of iodine for the different seasons to support this analysis, or at least highlight the competition between the two processes in determining the Monsoon influence on the iodine burden over the Indian ocean MBL.

a. Which specific iodine species suffer washout in the iodine chemical scheme, and how different are the modeled/observed precipitation regimes for the pre-monsoon and Monsoon seasons?

b. I've been also capable of finding a couple of reference to wet-deposition in the text (P14/L316; P25/L580), but only applies to Ozone and NOx. Equivalent descriptions relating the impact of wet-scavenging on iodine washout should be provided.

c. The flux strength in the model was reduced by 40%, but it is not explicitly informed how. Did you compute the flux strength at each model pixel using the Carpenter/MacDonald parameterization and then multiply it by 0.6? Just it? Would it be possible/worth to perform an equivalent simulation maintaining the source flux unaltered and increasing the washout by 40%?

**Minor Comments:**

P2/L27: It could be useful to provide a range of iodine values in the abstract.

P2/L31: Values provided in the abstract are maximum regional changes. It could be of use to mention in the abstract that mean values across the modeled domain are smaller.

P3/L41: I suggest replacing "implicated in" with "associated to".

P4/L65: "Until recently, the Indian Ocean was **one of** the most under-sampled region**s** for iodine species …"

P6/L113: I was surprised the authors did not mention at the end of the introduction that one of the main outcomes of this work was to adjust the iodine source parameterization to obtain a consistent model-observation validation.

P7/L142: "the drastic differences in air masses over the three seasons". I'm not sure if drastic is the proper adjective to use here, and it should be explicit mention that differences are on the "transport" of air masses.

P7/L145-149: It should be mentioned at some point that the bromine and chlorine chemistry scheme are identical for all simulations.

P8/L168: "discussed further **in** Section 3.2".

P8/L182-184: "The model captures well the difference between the IIOE-2 and the ISOE-8 cruises, which started from the west and east coasts of India, respectively". I do not see such a variation for ozone, mostly considering that Fig. 3 shows only 1 set of results for WRF-Chem output without distinguishing between cruises. Could you please explain? Having said this, are there any other ozone observations available (in addition to IIOE-2 and ISOE-8) to compare with model results for the remaining seasons? If no additional measurements are available, at least a comment on the text would help to support the presented implications for ozone.

P9/L196-197: "iodine chemistry would not have any measurable impact". Why would not instead of does not? You are pointing out to model results that allow to compute the impact.

P9/L204: "However, despite  being an area of high productivity …"; and leave "by a factor of 10-20 outside the brackets.

P13/L286: Avoid the excessive usage of "only"

P13/L293: "… rather than the photolysis of organoiodides, which are long-lived and hence do not contribute heavily to the MBL". I understand the authors are comparing the lifetimes of organic iodine species with respect to inorganic iodine species. But note that organoiodide species are usually referred in the literature as very short-lived species, to distinguish them from the long-lived CFCs and halons. Thus, I suggest rephrasing the sentences to avoid confusion.

P14/L323: "If only the MBL **is considered**, where elevated concentration of IO are observed,  …"

P15/L340: Make sure the minus sign sticks to the number within the same line

P15/L349-351: "The reason for larger absolute differences as compared to mean differences is that there are both increases and decreases seen through the domain, and hence the absolute differences gives us an idea of the total impact of iodine chemistry.". I do not understand the rationale for this type of analysis. See my major comment Nº2c.

P16/L371-373: "The fact that the absolute change values are close to the mean change values shows that most of the domain sees a destruction in ozone due to the presence of iodine compounds.". Similar to previous comment, I do not understand the rationale for this type of Analysis. If authors want to highlight that iodine-driven ozone destruction is larger than production, this is already shown in Fig. 5.

P16/L374-382: The different percentage impacts in comparison with other studies is well oriented and highlights the different chemical treatments between studies. However, I suggest including here an explicit mention to the fact that the iodine source parameterization has been reduced here, which clearly affect the percentage impact obtained.

P18/L414-415: Why the NOx abundance over the shipping lanes are more marked for NO than for NO2? P18/L418-419: Why January show the lowers NO concentration over the shipping lanes? Correct the typo on P18/L422 to make reference to NO instead of NO2.

P18/L424-426: The authors attribute the smaller standard deviation over the MBL to the "much cleaner air than above the Indian subcontinent". Could the smaller deviation also be related to the less pronounced day/night variability of dominant NOx shipping sources compared to continental NOx sources?

P19/L441: "Over the MBL too, …" Please rephrase.

P19/L446-448: However, similar to NO2, these values are misrepresentative of the effect of IO because of differences in the sign of the change across the domain.". I do not understand the rationale for this type of analysis. See my major comment Nº2c.

P20/L457 and P23/L536: shift the position of HAL and BASE to make it consistent with the percentage change computation.

P20/L458: "decreases in NOx as high as 50%...". Was this value computed for the model monthly mean? If that is the case, which are the maximum differences for a specific day or hour?

P20/L471-473: Using "slightly higher" is confusing, as comparatively, the percentage change in July is more than 5 times larger when only the MBL is considered. Also, note that for all this values, the standard deviation is much larger than the mean, highlighting the huge variability on the averaged values, so interpretation should be taken with caution. This should be highlighted in the text.

P20/L475-479 and P26/L603-605: I would expect that, in addition, differences between Li et al., 2019 and the present work are affected by the larger oceanic fraction of the model domain for the indian ocean study in comparison with the mostly continental European domain of Li et al.

P23/L543-545: The way the sentence is written seems to indicate positive differences when they are negative. I suggest rephrasing.

P23/L547-551: "The mean percentage change in the OH and HO2 mixing ratios peaks at 2.6 % and 8.4 % for the months of April and July, respectively (Table 2), but the absolute percentage change in OH is higher at 3.6 % in January, while the HO2 absolute percentage change (Table 3) is about ~8.4 % showing the large impact of iodine chemistry on the oxidation capacity of the MBL.". I do not understand the analysis and implications. See my major comment Nº2c.

P26/L607: Are 24 hs mean used or only night-time values considered?

P26/L610-612: Why the larger changes in NO3 are predicted during the period of time when iodine chemistry is less active? Is this a day/night issue?

**Figures and Tables:**

Fig. 1, 4-10: The indication of longitude in all panels appears in the middle of the domain and not at the axis, which makes it very difficult to read. Also note that all figures captions start with "Model simulations showing …" and ends with "… are shown". Replace lower panels by bottom panels. Please rephrase.

Fig. 2: It might be possible to include the cruise tracks on any of the panels of Fig. 1 to reduce the number of Figures.

Fig. 3: I get a confusion with the "empty squares" symbols for IO, as they are supposed to be "upper limits" but at 12ºN and 8ºN there is a whisker-range line also for larger values than the square.

---

## Referee Comment (RC2) · Rolf Sander (Referee) · 21 Jan 2021

Mahajan et al. present a very interesting modeling study about iodine chemistry in the mbl above the Indian Ocean. Although the manuscript contains several important results, I see two major problems: First, it is very long and tedious to read because of redundant information and commonplaces. Second, some important items should be explained in more detail or analyzed further. I recommend publication after major revisions. My suggestions are explained in more detail below:

1) SUGGESTIONS FOR REMOVING REDUNDANT AND LESS IMPORTANT PARTS OF THE MANUSCRIPT

[Figure]

- Tables 2 and 3 show differences, percentage differences, absolute differences, and absolute percentage differences between HAL and BASE. I don't think it is necessary to present 4 different ways to compare the scenarios. It is difficult for the reader to understand these quantities as the difference between "absolute changes" and "mean changes" is not defined.

- Don't repeat the numbers from the tables in the text. In most cases, it would be sufficient to refer to the tables.

- For a lot of the numbers, especially when comparing HAL to BASE, the standard deviation is larger than the value (e.g., l. 341: "a small increase of $0.01\pm0.31$ ppbv"). I suspect that such numbers are undistinguishable from zero. This makes it even less important to discuss their values in the text. In many cases, it may be sufficient to state that the value is not affected by iodine chemistry.

- There are many statements in the text describing quite obvious facts which are not even related to iodine chemistry. I suggest to remove them. A few examples are:

— l.306-307: "much higher concentrations of O3 are observed over the Indian subcontinent as compared to in the surrounding ocean MBL"

— l.392-393: "much higher concentrations of NO2 are observed over the Indian subcontinent as compared to the surrounding ocean MBL"

— l.396-398: "A sharp decrease is observed from the coast to the open ocean environment, which is expected considering that the primary sources of NO2 are on the subcontinent."

— l.407-408: "This shows that the MBL is much cleaner than the air above the Indian subcontinent."

— l.409-410: "NO also shows higher concentrations over the Indian subcontinent as compared to the surrounding ocean MBL"

— l.424-425: "the MBL is much cleaner than the air above the Indian subcontinent"

— l.501-504: "There is a correlation between the hotspots for NOx, and low concentrations of HO2 over the Indian subcontinent. This is due to the titration of HO2 by NO, which forms NO2 and leads to an increase in O3 formation."

2) IMPORTANT RESULTS THAT SHOULD BE EXPLAINED BETTER OR ANALYZED FURTHER

- The HAL scenario seems to contain the halogens I, Br, and Cl. However, it is not clear to me if the BASE scenario contains Cl and Br chemistry or if it is without halogens. This should be mentioned in the text.

- I think the most important result of this study is that the current parameterization for the inorganic iodine flux needs to be reduced. It is also mentioned (l. 248) that "models tend to underestimate the sources of nitrogen in the open ocean resulting in lower levels of NOx in the MBL". I suggest to make another model run with the full inorganic iodine flux and higher NOx to check if this can also produce realistic results for IO.

- l.387: "the resultant increase or decrease in nitrogen oxides depends on the concentrations of iodine compounds"

I don't understand how iodine compounds can increase nitrogen oxides. Please provide a chemical reaction to explain this. Decomposition of INO3 is not a real source, it only regenerates NO2 which was previously consumed in the formation of INO3.

- l.436-437: "In most of the shipping lanes, where high NO is observed, the inclusion of iodine chemistry leads to an increase in the NOx concentrations"

This is a very interesting result which should be investigated further! Which reactions in the model cause this effect? Or is this an effect of transport?

- l.482-483: "Hydrogen oxides are impacted by iodine chemistry through the catalytic

reaction involving IO changing HO2 into OH."

Is this really the main effect? Please compare this to the indirect effect when IO reduces O3, which in turn reduces the OH production from O3.

- l.594-595: "there are pockets of an increase in NO3 observed over the subcontinent."

This is another very interesting result! Why does iodine increase NO3 at some locations, and decrease NO3 at others? Is this just numerical noise, or is there an explanation for this?

3) MINOR COMMENTS

- l.44: "The known effects include [...] oxidation of mercury (Wang et al., 2014)"

I don't think that oxidation of mercury via iodine chemistry is established. Wang et al. investigated it based on theoretical calculations by Goodsite, and they came to the conclusion that NO2 and HO2 are more important for RGM generation.

- l.60: "concentrations reaching as high as ∼3 parts per trillion by volume (pptv)"

This is a mixing ratio, not a concentration.

Also, note that according to the IUPAC Recommendations (page 1387 of Schwartz & Warneck "Units for use in atmospheric chemistry", Pure & Appl. Chem., 67(8/9), 1377-1406, 1995, https://www.iupac.org/publications/pac/pdf/1995/pdf/6708x1377.pdf) the usage of "ppb" and "ppt" is discouraged for several reasons. Instead, "nmol/mol" and "pmol/mol" should be used for gas-phase mixing ratios. I suggest to replace these obsolete units.

- l.101: Change "Li et al. (Li et al., 2019)" to "Li et al. (2019)"

- l.173: Change "The levels observed and simulated IO" to "The levels OF observed and simulated IO"

- l.214-215: "...even when the uncertainty in the observations is considered (Figure 2)"

I cannot see the uncertainties in Figure 2.

- l.228: "The second reason for overestimating..."

It has already been shown that the first reason (seawater iodide concentrations) cannot explain the overestimation. Thus, it may be better to say "the second POTENTIAL reason for overestimating..."

- l.805: Add volume and page numbers to the reference Mahajan (2019b).

———————————————————

---

## Author Comment (AC1) · 23 Mar 2021

**Response to reviewer comments for manuscript number: acp-2020-1219**

Comments by reviewers are shown in italic typeface and the responses shown normal typeface.
* * *
Reviewer 1:

*Review of "Modelling the Impacts of Iodine Chemistry on the Northern Indian Ocean Marine Boundary Layer" by Mahajan et al., ACPD, 2020 The paper presents a computational evaluation of the impact of oceanic iodine emissions (both organic and inorganic) on the composition of the lower atmosphere, with a special focus on the changes on ozone, HOx and NOx (plus NO3) over the Indian ocean and sub-continent. The most relevant results are: i) the requirement to reduce by 40% the strength of the oceanic emissions of inorganic iodine in comparison with previous modeling studies to properly reproduce IO observations performed over the Indian ocean; and ii) the different seasonal impact of iodine chemistry predicted during the pre-monsoon and monsoon periods, mostly due to the changes on the total iodine burden during the different seasons. The paper is well organized and the results are presented in order, although in many sections the writing style, figures format and absolute/percentage change description is repetitive, without a comprehensive interpretation on how most of the modeled changes for each individual species are correlated with the others. This work is certainly of interest and fulfills the requirements and objectives of Atmospheric Chemistry and Physics. However, major changes must be performed before acceptance for publication as described below.*
RESPONSE: We thank the reviewer for the positive comments on the manuscript and have made changes according to the specific points that have been raised below.

*Major Comments:*

*1. Model Validation:*
*The paper organization and results presentation is correct, starting with the model validation and followed by a model analysis. However, many technical details are missing (both for measurements data and model output processing), which prevents the reader to understand how well the model compares with observations. For example:*

*a. How the cruise observations were averaged? Every 4° Latitude? Every 2° Latitude? By looking at Figure 2 (ship-tracks) and Figure 3 (IO and O3 vmr), it is evident that IIOE-2 and ISOE-8 observations were post-processed in a different way, but no details are given. Which is the temporal resolution of the observations, and how many days of measurements were considered?*
RESPONSE: The observations for the cruises were averaged daily so that the diurnal variation in IO was not a driving factor for any observed differences. This is now explicitly mentioned in the paper: 'daily averaged' (Line 159). The processing of the data was done in the same way for the two campaigns but looks different due to the ship speeds. Data processing details are presented in previously published papers, as cited in the manuscript.

*b. How the model output was extracted and processed to compare with observations? As can be seen in Fig. 3, only 1 dataset of model output is compared with 2 independent cruises. As*

*IIOE-2 (Arabian sea) and ISOE-8 (bay of Bengal) campaigns were performed in different seas (and many of the results presented in Figs. 4-10 highlight differences among these regions), wouldn't it be better to compare model vs. observations for each cruise independently?*
RESPONSE: The ship location data was used for extracting the model output for the days where we had observations and then their mean was taken. So, each datapoint is representative of the individual cruises and hence accounts for the different regions. This is now made clear in the caption of the Figure 3. 'A comparison between the model simulated and observed daily averaged IO (top panel) and $O_3$ (bottom panel) as per the cruise locations is shown. In cases where IO was not detected, an upper limit (empty squares – the errors on the empty squares show the range of upper limits for that day) was estimated. The model validation was performed only for the winter period, when the cruise-based data was available.'

*c. Did you extract only the closest model pixels to the location of the cruise or did you average a larger region surrounding the ship-lane and/or the whole oceanic domain? Did you consider the whole monthly mean model output to compare with observations, or did you use only a small time-window equivalent to the one for observations? In case the former option, how is the model variability between the monthly mean and the observations-window time period?*
RESPONSE: Yes, the closest model pixel to the location of the ships was chosen for the model evaluation. We considered the model output within the same period as the ship observations. This is now made clear in the manuscript as per the response to the comment above.

*d. The paper presents a model analysis for three different seasons (pre-monsoon in April, summer monsoon in July and winter monsoon in January), while model validation is performed only for winter monsoon when IIOE-2 (December) and ISOE8 (January) cruises took place. Even when this can be inferred from the text, I suggest explicitly pointing at the lacking of a complete seasonal validation of model results within the model validation section as well as in the Figure caption and discussion. Additionally, it would be useful to provide a clear statement within the conclusions that the addressed model impacts for the remaining seasons should be taken with caution, as none the ozone nor the IO results were validated during those seasons.*
RESPONSE: We have now added a statement regarding the lack of validation for all the seasons, as requested: 'The model was validated with observations from two cruises during the winter season.' (Line 578 and Figure 3 caption).

*2. Modeled Geographical Distributions and Statistics:*

*Once the model is validated, a series of Figures, Tables and Values describing the impact of HAL vs. BASE simulation is provided. Even when in general the presentation is consistent, many of the results are repetitive and presented as a consecutive sequence of absolute and percentage impacts, distinguishing between the whole domain and when a landmask is applied, and written using almost the same style. I believe this description can be improved, making the text more easy-going, pointing at the proper table/figure when results are presented, and mostly by describing absolute/relative impacts altogether as well as making inter-connections between the changes on the abundances of the different species analyzed. In particular:*

*a. Many paragraphs of sections 3.4 (NOx), 3.5 (HOx) and 3.6 (NO3) start with "The middle panels of Fig. # show the absolute difference ..." and/or "The bottom panels in Fig. # shows*

*the percentage change …". Many portions of this sections present a large series of model mean values ± standard deviation that can be found in Tables 2 and 3, and in some cases no additional interpretation to the number values is provided. Even when this style can be helpful at the beginning of the results description to orientate the reader on how Figures and Tables should be read, as the paper moves forward it would be very useful not to repeat the full list of values on the text, but just use those that are necessary to justify the analysis being described.*
RESPONSE: We have now shortened the text to remove the repetitive sections.

*b. Something similar is present when mean results above the MBL are compared with those over the Indian-subcontinent or the whole model domain. Quantitative results are literally copied from the table into the text for all species and simulations, and usually equivalent conclusions are provided. I found this could be largely reduced to avoid unnecessary repetition, and only include this type of explicit analysis when a special issue must be highlighted, leaving for the remaining of the text a more general comment an interconnection to the results for other species.*
RESPONSE: We have now removed the repetition of the values which appear in the tables.

*c. In P15/L345-346 and Table 3 (as well as in other sections below), the "absolute changes rather than the mean changes" are presented. The exact difference between each of the magnitudes should be clearly described at first usage. My understanding is that the "mean change" is the signed difference (HAL-BASE), while the "absolute change" is the unsigned module-difference |HAL-BASE|. If that is the case, I do not completely understand the rationale for providing the unsigned difference, as for most of the results presented here using a signed analysis that allows determining a positive or negative deviation (bias) with respect to a base simulation would be sufficient. If the authors are interested in presenting statistical evaluation of the model performance (comparing HAL sensitivity respect to the BASE simulation), other statistical measures such as the Normalized Fractional Error (NME) or root mean squared error (RMSE) could be used (see Willmott, 1981. On the validation of models. Phys. Geogr. 2, 184–194.https://doi.org/10.1080/02723646.1981.10642213).*
RESPONSE: We have now clearly defined the terms 'difference' and 'absolute difference' (Line 341 and Line 357). The NME and RMSE terms can be useful for identifying differences but are not useful to identify the total impact on the composition and the total impact because of a species and hence we have not included them. But we have moved the absolute differences table to the supplementary text and simplified the discussion in several places to avoid confusion.

*d. Model results are provided for the mean volume mixing ratio within the lower 10 layers within the MBL (P7L156) instead of the surface mean. Even when I'm fine with using this procedure as it allows addressing a more realistic indication of the atmospheric impacts of introducing the organic and inorganic iodine sources on the model, it would be great if the authors could provide an indication of how variable is the vertical distribution within the MBL. For example, all values in the Tables and text present the standard deviation of the model results, but those values are more representative of the "spatial" averaging than the "vertical" averaging. Authors should decide whereas it is worth including an additional figure showing vertical profile or latitudinal cross-section across each of the cruises, but at least an indication of the magnitude of the vertical changes within the MBL would be useful.*

RESPONSE: The main reason for using a boundary layer average is that the measurements of iodine compounds are averaged across the boundary layer. Hence, to study the spatial variation, we have stuck to the parameters that we could validate. We have now included a figure on the vertical distribution in the supplementary text as requested (Figure S3) and added a brief discussion on this (Line 312): 'Figure S3 shows the vertical distribution of IO as simulated in the HAL scenario for the three months in the region between 2-10 °N and 65-73 °E close to the equator where the higher values are observed. Most of the IO is restricted to the boundary layer, with the IO mixing ratio reducing to less than a $10^{th}$ of the value of the surface above the boundary layer. This indicates that most of the iodine in the model is indeed from short lived gases, HOI and $I_2$ being the primary source'

*3. Iodine Oceanic Source vs Iodine wet deposition: The authors attribute the strong seasonal variation only to the seasonal change in iodine sources driven by the cleaner oceanic air (lower ozone) during summer Monsoon (P11L258-P12/L266). However, they do not provide any quantitative estimation of the overall source of HOI and I2 from the ocean during each season, presenting only changes in the IO vmr within the MBL. Even though large seasonal changes on emissions are expected, the Monsoon drives also large changes in precipitation, which in turn will impact on the washout/wet-scavenging of soluble species (and many iodine species are soluble). The authors should be able to quantify the net change in both the fluxes and also the sinks of iodine for the different seasons to support this analysis, or at least highlight the competition between the two processes in determining the Monsoon influence on the iodine burden over the Indian ocean MBL.*

RESPONSE: We have now included figures of the $I_2$ and HOI emissions to help with the identifying of the differences between the months (Figure S2) and discussed this in the manuscript: 'The main reason for this season change is the emissions of HOI and $I_2$ rather than due to deposition effects, as we can see significant changes in the HOI and $I_2$ emissions also during July (Figure S2)' (Line 279). The effect of wet scavenging is answered in the comment below.

a. Which specific iodine species suffer washout in the iodine chemical scheme, and how different are the modeled/observed precipitation regimes for the pre-monsoon and Monsoon seasons?

RESPONSE: The iodine species that undergo washout were described in detail in Badia et al. (2019) and the reference therein. The species include $INO_3$, HOI, HI, IBr, ICl, $INO_2$, $I_2$, $I_2O_2$, $I_2O_3$, and $I_2O_4$. We have added the relevant information in the model description section: 'Iodine species, including $INO_3$, HOI, HI, IBr, ICl, $INO_2$, $I_2$, $I_2O_2$, $I_2O_3$, and $I_2O_4$, undergo washout process as described in Badia et al. (2019) and the reference therein' (Line 133).

The simulated washout of atmospheric compositions (here taking $O_3$ as an example) in these three months (Jan, Apr, and Jul) are shown in Figure S1. The washout intensity is lower in the pre-monsoon season (Apr) and higher in the monsoon seasons (Jan and Jul). (Figure S1 and Line 136)

*b. I've been also capable of finding a couple of reference to wet-deposition in the text (P14/L316; P25/L580), but only applies to Ozone and NOx. Equivalent descriptions relating the impact of wet-scavenging on iodine washout should be provided.*

RESPONSE: We have added a sentence describing the washout in the revised text. Please refer to the response above.

*c. The flux strength in the model was reduced by 40%, but it is not explicitly informed how. Did you compute the flux strength at each model pixel using the Carpenter/MacDonald parameterization and then multiply it by 0.6? Just it? Would it be possible/worth to perform an equivalent simulation maintaining the source flux unaltered and increasing the washout by 40%?*

RESPONSE: Indeed, we reduced the emission fluxes of HOI and $I_2$ via multiplying them by 0.6. While we cannot rule out the potential uncertainty in the wet deposition module, we find it more plausible that parameterization of the HOI and $I_2$ emission flux used in the present work has some uncertainty, at least in the Indian Ocean, as discussed in detail in the main text. Running another case to examine the potential uncertainty of the wet deposition module is out of the scope of the present work. We have added a sentence in section 3.2 of the revised manuscript to introduce the possible uncertainty of wet deposition scheme in affecting the simulated abundance of iodine species: 'Potential uncertainty in other model configurations, e.g., washout process, could also lead to uncertainties in the simulated abundance of iodine species.' (Line 138).

Minor Comments:

*P2/L27: It could be useful to provide a range of iodine values in the abstract.*
RESPONSE: Added.

*P2/L31: Values provided in the abstract are maximum regional changes. It could be of use to mention in the abstract that mean values across the modeled domain are smaller.*
RESPONSE: Added.

*P3/L41: I suggest replacing "implicated in" with "associated to".*
RESPONSE: Changed.

*P4/L65: "Until recently, the Indian Ocean was one of the most under-sampled regions for iodine species ..."*
RESPONSE: Changed.

*P6/L113: I was surprised the authors did not mention at the end of the introduction that one of the main outcomes of this work was to adjust the iodine source parameterization to obtain a consistent model-observation validation.*
RESPONSE: We felt this was not needed in the introduction as it was an outcome of this study. It is included in the conclusions.

*P7/L142: "the drastic differences in air masses over the three seasons". I'm not sure if drastic is the proper adjective to use here, and it should be explicit mention that differences are on the "transport" of air masses.*
RESPONSE: Corrected.

*P7/L145-149: It should be mentioned at some point that the bromine and chlorine chemistry scheme are identical for all simulations.*
RESPONSE: Added (Line 121).

*P8/L168: "discussed further in Section 3.2".*
RESPONSE: Corrected.

*P8/L182-184: "The model captures well the difference between the IIOE-2 and the ISOE-8 cruises, which started from the west and east coasts of India, respectively". I do not see such a variation for ozone, mostly considering that Fig. 3 shows only 1 set of results for WRF-Chem output without distinguishing between cruises. Could you please explain? Having said this, are there any other ozone observations available (in addition to IIOE-2 and ISOE-8) to compare with model results for the remaining seasons? If no additional measurements are available, at least a comment on the text would help to support the presented implications for ozone.*
RESPONSE: The different cruises are colour coded in Figure 3 and the WRF-Chem output are extracted according to the location of the observations as explained before. This is now made clear in the caption.

*P9/L196-197: "iodine chemistry would not have any measurable impact". Why would not instead of does not? You are pointing out to model results that allow to compute the impact.*
RESPONSE: We corrected the word to make it clear.

*P9/L204: "However, despite the being an area of high productivity ..."; and leave "by a factor of 10-20 outside the brackets. P13/L286: Avoid the excessive usage of "only"*
RESPONSE: Corrected.

*P13/L293: "... rather than the photolysis of organoiodides, which are long-lived and hence do not contribute heavily to the MBL". I understand the authors are comparing the lifetimes of organic iodine species with respect to inorganic iodine species. But note that organoiodide species are usually referred in the literature as very short-lived species, to distinguish them from the long-lived CFCs and halons. Thus, I suggest rephrasing the sentences to avoid confusion.*
RESPONSE: Corrected.

*P14/L323: "If only the MBL is considered, where elevated concentration of IO are observed, is considered ..."*
RESPONSE: Corrected.

*P15/L340: Make sure the minus sign sticks to the number within the same line*
RESPONSE: Corrected

*P15/L349-351: "The reason for larger absolute differences as compared to mean differences is that there are both increases and decreases seen through the domain, and hence the absolute differences gives us an idea of the total impact of iodine chemistry.". I do not understand the rationale for this type of analysis. See my major comment Nº2c.*

RESPONSE: We have defined the terms clearly and simplified this discussion at several points in the revised paper to avoid confusion.

*P16/L371-373: "The fact that the absolute change values are close to the mean change values shows that most of the domain sees a destruction in ozone due to the presence of iodine compounds.". Similar to previous comment, I do not understand the rationale for this type of Analysis. If authors want to highlight that iodine-driven ozone destruction is larger than production, this is already shown in Fig. 5.*
RESPONSE: Yes, this is what we wanted to show. We have changed the discussion to avoid confusion.

*P16/L374-382: The different percentage impacts in comparison with other studies is well oriented and highlights the different chemical treatments between studies. However, I suggest including here an explicit mention to the fact that the iodine source parameterization has been reduced here, which clearly affect the percentage impact obtained.*
RESPONSE: Added.

*P18/L414-415: Why the NOx abundance over the shipping lanes are more marked for NO than for NO2?*
RESPONSE: This is due to the range of values, and hence relatively they are more noticeable, but we have changed this sentence to make it clear that it is not about more NO than $NO_2$.

*P18/L418-419: Why January show the lowers NO concentration over the shipping lanes?*
RESPONSE: This is due to the higher level of $O_3$ along the shipping lanes in January which destroys NO and $NO_2$.

*Correct the typo on P18/L422 to make reference to NO instead of NO2.*
RESPONSE: Corrected.

*P18/L424-426: The authors attribute the smaller standard deviation over the MBL to the "much cleaner air than above the Indian subcontinent". Could the smaller deviation also be related to the less pronounced day/night variability of dominant NOx shipping sources compared to continental NOx sources?*
RESPONSE: Good point – however considering the much lower mean value, the contribution of shipping emissions is not large when integrated over the entire MBL.

*P19/L441: "Over the MBL too, …" Please rephrase.*
RESPONSE: Changed.

*P19/L446-448: However, similar to NO2, these values are misrepresentative of the effect of IO because of differences in the sign of the change across the domain.". I do not understand the rationale for this type of analysis. See my major comment Nº2c.*
RESPONSE: As mentioned in the comments regarding the absolute difference, we have changed the entire discussion when discussing the signs.

*P20/L457 and P23/L536: shift the position of HAL and BASE to make it consistent with the percentage change computation.*
RESPONSE: Changed.

*P20/L458: "decreases in NOx as high as 50%...". Was this value computed for the model monthly mean?*
RESPONSE: This is the difference in the monthly mean.

*P20/L471-473: Using "slightly higher" is confusing, as comparatively, the percentage change in July is more than 5 times larger when only the MBL is considered. Also, note that for all this values, the standard deviation is much larger than the mean, highlighting the huge variability on the averaged values, so interpretation should be taken with caution. This should be highlighted in the text.*
RESPONSE: We have changed the phrase and added a sentence about the large standard deviations in the text at several places.

*P20/L475-479 and P26/L603-605: I would expect that, in addition, differences between Li et al., 2019 and the present work are affected by the larger oceanic fraction of the model domain for the indian ocean study in comparison with the mostly continental European domain of Li et al.*
RESPONSE: This is correct, which is why we have also computed the 'only MBL' fraction in the manuscript.

*P23/L543-545: The way the sentence is written seems to indicate positive differences when they are negative. I suggest rephrasing.*
RESPONSE: Rewritten.

*P23/L547-551: "The mean percentage change in the OH and HO2 mixing ratios peaks at 2.6 % and 8.4 % for the months of April and July, respectively (Table 2), but the absolute percentage change in OH is higher at 3.6 % in January, while the HO2 absolute percentage change (Table 3) is about ~8.4 % showing the large impact of iodine chemistry on the oxidation capacity of the MBL.". I do not understand the analysis and implications. See my major comment N°2c.*
RESPONSE: We have removed the confusing discussion as detailed in the earlier comment.

*P26/L607: Are 24 hs mean used or only night-time values considered? P26/L610-612: Why the larger changes in NO3 are predicted during the period of time when iodine chemistry is less active? Is this a day/night issue?*
RESPONSE: The values presented are 24-hour averages. However, this should not be a day-night issue as one of the main inorganic emissions, $I_2$, reacts with $NO_3$ and is active during the night-time too. Keeping 24-hour averages helps in a direct comparison with the other parameters.

Figures and Tables:

*Fig. 1, 4-10: The indication of longitude in all panels appears in the middle of the domain and not at the axis, which makes it very difficult to read. Also note that all figures captions start with "Model simulations showing …" and ends with "… are shown". Replace lower panels by bottom panels. Please rephrase.*

RESPONSE: Yes, the shape of the domain means that we would need labels on the top and bottom, because of which we thought that it is better to have the labels on the plot. We have changed the caption to avoid repetition. We have also replaced the 'lower panels' with 'bottom panels'.

*Fig. 2: It might be possible to include the cruise tracks on any of the panels of Fig. 1 to reduce the number of Figures.*

RESPONSE: We feel that adding the cruise tracks in the figure showing the winds would be too crowded and are happy to keep an extra figure.

*Fig. 3: I get a confusion with the "empty squares" symbols for IO, as they are supposed to be "upper limits" but at 12ºN and 8ºN there is a whisker-range line also for larger values than the square.*

RESPONSE: Yes, that is the standard deviation of the upper limits for that day – this is due to the measurements using the DOAS method, in which the upper limits change through the day. This is now made clear in the caption.
* * *
Reviewer 2:

*Mahajan et al. present a very interesting modeling study about iodine chemistry in the mbl above the Indian Ocean. Although the manuscript contains several important results, I see two major problems: First, it is very long and tedious to read because of redundant information and commonplaces. Second, some important items should be explained in more detail or analyzed further. I recommend publication after major revisions. My suggestions are explained in more detail below:*

RESPONSE: We thank the reviewer for the positive comments and have made significant changes as per the comments below.

1) SUGGESTIONS FOR REMOVING REDUNDANT AND LESS IMPORTANT PARTS OF THE MANUSCRIPT

*Tables 2 and 3 show differences, percentage differences, absolute differences, and absolute percentage differences between HAL and BASE. I don't think it is necessary to present 4 different ways to compare the scenarios. It is difficult for the reader to understand these quantities as the difference between "absolute changes" and "mean changes" is not defined.*

RESPONSE: Yes, we had also given a great deal of thought to this. However, considering the changes are positive and negative, we felt that both mean and absolute means were necessary. The mean gives the total impact, but the absolute change also helps us understand how important iodine chemistry is. That being said, we have now moved the table with the absolute changes into the supplementary text and defined it clearly in the paper in addition to simplifying the discussion in several places.

*- Don't repeat the numbers from the tables in the text. In most cases, it would be sufficient to refer to the tables.*
RESPONSE: We have removed the repeated numbers.

*- For a lot of the numbers, especially when comparing HAL to BASE, the standard deviation is larger than the value (e.g., l. 341: "a small increase of 0.01±0.31 ppbv"). I suspect that such numbers are undistinguishable from zero. This makes it even less important to discuss their values in the text. In many cases, it may be sufficient to state that the value is not affected by iodine chemistry.*
RESPONSE: Rather than it not being important, it shows the problems with using averages, when the variance is larger. It basically means that in certain regions, the value is highly affected. We have changed this discussion at multiple points in the text to highlight this.

*-There are many statements in the text describing quite obvious facts which are not even related to iodine chemistry. I suggest to remove them. A few examples are:*

> *— l.306-307: "much higher concentrations of O3 are observed over the Indian subcontinent as compared to in the surrounding ocean MBL"*
> RESPONSE: Removed

> *— l.392-393: "much higher concentrations of NO2 are observed over the Indian subcontinent as compared to the surrounding ocean MBL"*
> RESPONSE: Removed.

> *— l.396-398: "A sharp decrease is observed from the coast to the open ocean environment, which is expected considering that the primary sources of NO2 are on the subcontinent."*
> RESPONSE: Removed

> *— l.407-408: "This shows that the MBL is much cleaner than the air above the Indian subcontinent."*
> RESPONSE: Removed.

> *— l.409-410: "NO also shows higher concentrations over the Indian subcontinent as compared to the surrounding ocean MBL"*
> RESPONSE: Removed.

> *— l.501-504: "There is a correlation between the hotspots for NOx, and low concentrations of HO2 over the Indian subcontinent. This is due to the titration of HO2 by NO, which forms NO2 and leads to an increase in O3 formation."*
> RESPONSE: Removed.

*2) IMPORTANT RESULTS THAT SHOULD BE EXPLAINED BETTER OR ANALYZED FURTHER*

*- The HAL scenario seems to contain the halogens I, Br, and Cl. However, it is not clear to me if the BASE scenario contains Cl and Br chemistry or if it is without halogens. This should be mentioned in the text.*

RESPONSE: Yes, the only change was the exclusion of iodine chemistry. This is now mentioned clearly in the model setup section.

*- I think the most important result of this study is that the current parameterization for the inorganic iodine flux needs to be reduced. It is also mentioned (l. 248) that "models tend to underestimate the sources of nitrogen in the open ocean resulting in lower levels of NOx in the MBL". I suggest to make another model run with the full inorganic iodine flux and higher NOx to check if this can also produce realistic results for IO.*

RESPONSE: We agree that this is one of the most important results and have highlighted this in the abstract and conclusions. Regarding another model run for the iodine-$NO_x$ interaction, we would be happy to do one if we had any measure of the $NO_x$ levels in the MBL. However, considering that we have no observations for validation and the fact that $NO_x$ levels are expected to be lower than most commonly used instrument detection limits in the open ocean, it is difficult to decide how much to increase the $NO_x$ by for a sensitivity test. Increasing $NO_x$ indefinitely will eventually reproduce the IO, but without having any constraints on the $NO_x$ concentrations, we do not feel more simulations are warranted. Additionally, an increase in $NO_x$ will increase the ozone concentrations, which are already overestimated. An increase in $O_3$ will, in turn, increase the emission flux of iodine species and further increase the level of IO. We have added a brief discussion about this in the paper: 'Observations of $NO_x$ in the MBL are rare given the low concentrations, and no observations have been made in the Indian Ocean MBL. However, an increase in $NO_x$ would lead to an increase in the ozone, which is already slightly overestimated in the model' (Line 261)

*- l.387: "the resultant increase or decrease in nitrogen oxides depends on the concentrations of iodine compounds" I don't understand how iodine compounds can increase nitrogen oxides. Please provide a chemical reaction to explain this. Decomposition of INO3 is not a real source, it only regenerates NO2 which was previously consumed in the formation of INO3.*

RESPONSE: This has been answered in the comment below.

*- l.436-437: "In most of the shipping lanes, where high NO is observed, the inclusion of iodine chemistry leads to an increase in the NOx concentrations" This is a very interesting result which should be investigated further! Which reactions in the model cause this effect? Or is this an effect of transport?*

RESPONSE: Along the shipping lanes, the level of $NO_x$ is mostly controlled by the levels of $O_3$, OH, and $HO_2$, which were reduced by the addition of iodine chemistry (mainly through the reaction of I+$O_3$->IO and the subsequent effects on $HO_x$ levels). Therefore, the loss of $NO_x$ through $O_3$, OH, and $HO_2$ were decreased resulting in an increase of NO and $NO_2$. We have added the following sentence in the revised manuscript: 'The addition of iodine chemistry reduces the level of $O_3$, OH, and $HO_2$ along the shipping lanes, therefore the loss of $NO_x$ through its reaction with $O_3$, OH, and $HO_2$ is also reduced, resulting in an increase in the level of $NO_x$ along the shipping lanes.'

*- l.482-483: "Hydrogen oxides are impacted by iodine chemistry through the catalytic reaction involving IO changing HO2 into OH." Is this really the main effect? Please compare this to the indirect effect when IO reduces O3, which in turn reduces the OH production from O3.*

RESPONSE: Figure 8 and Figure 9 show the changes of OH and $HO_2$ due to the addition of iodine chemistry, in which the level of OH is increased in much of the oceanic area while that of $HO_2$ is mostly reduced. The reaction of IO with $O_3$ reduces the level of $O_3$ and therefore OH and $HO_2$. If this reaction was the main driver, the changes of OH and $HO_2$ should both decrease

in similar locations as $O_3$. However, this is not the case indicating that this is not the main effect. Therefore, we attribute the changes in OH and $HO_2$ in the majority of the domain mostly to the catalytic reaction involving IO changing $HO_2$ into OH.

*- l.594-595: "there are pockets of an increase in NO3 observed over the subcontinent." This is another very interesting result! Why does iodine increase NO3 at some locations, and decrease NO3 at others? Is this just numerical noise, or is there an explanation for this? 3)*
RESPONSE: The production of $NO_3$ is controlled by the level of $NO_2$ and $O_3$. Iodine chemistry induces both negative and positive changes in $NO_2$ and $O_3$, leading to positive changes in $NO_3$ in some places, and negative in others.

**MINOR COMMENTS**

*- l.44: "The known effects include [...] oxidation of mercury (Wang et al., 2014)" I don't think that oxidation of mercury via iodine chemistry is established. Wang et al. investigated it based on theoretical calculations by Goodsite, and they came to the conclusion that NO2 and HO2 are more important for RGM generation.*
RESPONSE: We have rephrased this.

*- l.60: "concentrations reaching as high as ~3 parts per trillion by volume (pptv)" This is a mixing ratio, not a concentration. Also, note that according to the IUPAC Recommendations (page 1387 of Schwartz & Warneck "Units for use in atmospheric chemistry", Pure & Appl. Chem., 67(8/9), 1377- 1406, 1995, https://www.iupac.org/publications/pac/pdf/1995/pdf/6708x1377.pdf) the usage of "ppb" and "ppt" is discouraged for several reasons. Instead, "nmol/mol" and "pmol/mol" should be used for gas-phase mixing ratios. I suggest to replace these obsolete units.*
RESPONSE: Yes, although this was discouraged by IUPAC more than two decades ago, ppt and ppb are still commonly used in atmospheric chemistry. That being said, we have defined these as suggested by the reviewer.

*- l.101: Change "Li et al. (Li et al., 2019)" to "Li et al. (2019)"*
RESPONSE: Corrected.

*– l.173: Change "The levels observed and simulated IO" to "The levels OF observed and simulated IO"*
RESPONSE: Corrected.

*- l.214-215: "...even when the uncertainty in the observations is considered (Figure 2)" I cannot see the uncertainties in Figure 2.*
RESPONSE: Changed to Figure 3.

*- l.228: "The second reason for overestimating..." It has already been shown that the first reason (seawater iodide concentrations) cannot explain the overestimation. Thus, it may be better to say "the second POTENTIAL reason for overestimating..."*
RESPONSE: Corrected.

*- l.805: Add volume and page numbers to the reference Mahajan (2019b)*
RESPONSE: Added.